# Beyond Linear Approximations: A Novel Pruning Approach for Attention Matrix

**Yingyu Liang**[*]    **Jiangxuan Long**[†]    **Zhenmei Shi**[‡]    **Zhao Song**[§]    **Yufa Zhou**[¶]

## Abstract

Large Language Models (LLMs) have shown immense potential in enhancing various aspects of our daily lives, from conversational AI to search and AI assistants. However, their growing capabilities come at the cost of extremely large model sizes, making deployment on edge devices challenging due to memory and computational constraints. This paper introduces a novel approach to LLM weight pruning that directly optimizes for approximating the attention matrix, a core component of transformer architectures. Unlike existing methods that focus on linear approximations, our approach accounts for the non-linear nature of the Softmax attention mechanism. We provide theoretical guarantees for the convergence of our Gradient Descent-based optimization method to a near-optimal pruning mask solution. Our empirical results demonstrate the effectiveness of our non-linear pruning approach in maintaining model performance while significantly reducing computational costs, which is beyond the current state-of-the-art methods, i.e., SparseGPT and Wanda, by a large margin. This work establishes a new theoretical foundation for pruning algorithm design in LLMs, potentially paving the way for more efficient LLM inference on resource-constrained devices.

## 1 Introduction

Large Language Models (LLMs) based on the transformer architecture (Vaswani et al., 2017), including GPT-4o (OpenAI, 2024a), Claude (Anthropic, 2024), and OpenAI's recent o1 (OpenAI, 2024b), have shown immense potential to enhance our daily lives. They revolutionize fields like conversational AI (Liu et al., 2024), AI agents (Xi et al., 2023; Chen et al., 2024b), search AI (OpenAI, 2024b), and AI assistants (Mahmood et al., 2023; Zhang et al., 2023a; Kuo et al., 2024; Feng et al., 2024). With their growing capabilities, LLMs are powerful tools shaping the future of technology. However, the current state-of-the-art LLM weights number is extremely large. For instance, the smallest version of Llama 3.1 (Llama Team, 2024) needs 8 billion parameters, which takes more than 16GB GPU memory with half float precision and requires significant inference time. Deploying such models on edge devices such as smartphones becomes challenging due to their large memory and high computational cost. To reduce the LLM model size, many studies work on pruning the LLMs model weights to relax the device memory constraint and minimize response latency. The classical pruning problem in LLMs can be formulated as follows. Given a weight matrix $W \in \mathbb{R}^{d \times d}$ and some calibration data $X \in \mathbb{R}^{n \times d}$, where $n$ is input token length, and $d$ is feature dimension, the goal is to find a matrix $\widetilde{W}$ under some sparse constraint such that $\|XW - X\widetilde{W}\|$ being small under some norm function. The above formulation has been widely used in many state-of-the-art pruning methods, such as SparseGPT (Frantar & Alistarh, 2023) and Wanda (Sun et al., 2024).

However, the current object functions only focus on the approximation of a linear function $XW$. Their optimal solutions do not have a good approximation to the attention matrix (see Figure 2 for details). Note that the attention mechanism is the kernel module of the transformer architecture. The high-level insight of their bad performance is that the Softmax function is very sensitive to

---

[*] `yingyul@hku.hk`. The University of Hong Kong.    `yliang@cs.wisc.edu`. University of Wisconsin-Madison.

[†] `lungchianghsuan@gmail.com`. South China University of Technology.

[‡] `zhmeishi@cs.wisc.edu`. University of Wisconsin-Madison.

[§] `magic.linuxkde@gmail.com`. The Simons Institute for the Theory of Computing at UC, Berkeley.

[¶] `yufazhou@seas.upenn.edu`. University of Pennsylvania.

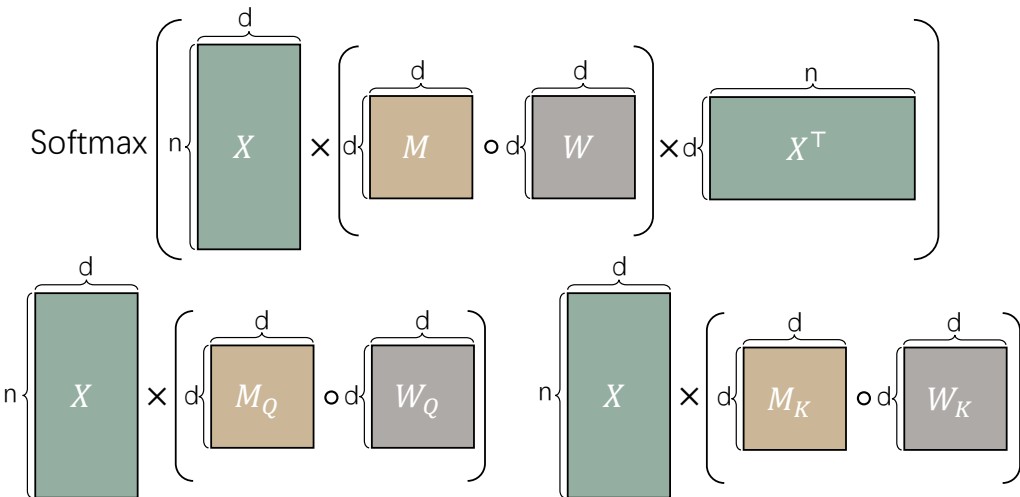

Figure 1: Comparison of our Attention Weights Pruning method and Linear Pruning method such as Wanda and SparseGPT. The top figure illustrates our proposed method of the attention matrix approximation, where pruning is applied directly to the fused attention weight matrix $W$, using only one pruning mask $M$. The bottom figure describes the Linear Pruning method of the linear function approximation, where pruning is applied separately to the query weight matrix $W_Q$ and key weight matrix $W_K$, using two different pruning masks $M_Q$ and $M_K$, respectively.

the large positive values of the input due to its $\exp$ scaling effect, while pruning mask based on linear approximation cannot capture this sensitivity. Thus, in this work, we directly compute the pruning mask on weights to approximate the attention matrix, which is a highly non-linear function, $\mathsf{Softmax}(XWX^\top) \in \mathbb{R}^{n \times n}$. To the best of our knowledge, this paper is the first work studying attention weight pruning to directly approximate the attention matrix. We provide a theoretical guarantee that optimization based on Gradient Descent (GD) on our loss function can converge to a good pruning mask solution (Theorem 1.3). Furthermore, we preliminarily verified the effectiveness of our method with empirical support (Section 6). Our theoretical foundation may pave the way for more efficient LLM inference on resource-constrained devices.

**Key background.** We introduce some key backgrounds. We define the attention matrix in the self-attention mechanism as below:

**Definition 1.1** (Attention Matrix). *Let $X \in \mathbb{R}^{n \times d}$ be the input. Given query and key weights matrix $W_Q, W_K \in \mathbb{R}^{d \times d}$, we define $W := W_Q W_K^\top$. Then, we have the Softmax attention matrix being*

$$\mathsf{Softmax}(XWX^\top) = D^{-1} \exp(XWX^\top),$$

*where (1) $D := \mathrm{diag}(\exp(XWX^\top) \cdot \mathbf{1}_n)$, (2) $\exp$ denotes the exponential function and is applied entry-wisely, (3) $\mathrm{diag}()$ operation takes a vector and outputs a diagonal matrix with the entries of that vector, and (4) $\mathbf{1}_n$ denotes the length-$n$ all ones vector.*

Further, we introduce the problem setup of our Attention Weights Pruning. By selectively reducing the number of non-zero elements in the attention weight matrix $W$ in Definition 1.1, we can preserve model performance while lowering computational cost and GPU memory usage. Below, we formally define the Attention Weights Pruning problem and the corresponding loss function:

**Definition 1.2** (Attention Weights Pruning). *Let $M \in [0,1]^{d \times d}$ be the pruning mask. Let $X, W$ be defined in Definition 1.1. Let $A := \exp(XWX^\top)$ and $\widetilde{A} := \exp(X(M \circ W)X^\top)$, where $\circ$ is the Hadamard product. Let $D := \mathrm{diag}(A \cdot \mathbf{1}_n)$ and $\widetilde{D} := \mathrm{diag}(\widetilde{A} \cdot \mathbf{1}_n)$. Let $\lambda \in \mathbb{R}_+$ be the regularization parameter. We define the Attention Weights Pruning loss function to be*

$$\mathcal{L}(M) := \frac{1}{2}\|D^{-1}A - \widetilde{D}^{-1}\widetilde{A}\|_F^2 + \frac{1}{2}\lambda\|M\|_F^2.$$

*Thus, the Attention Weights Pruning optimization problem is $\min_M \mathcal{L}(M)$.*

**Our contributions.** This is the first work studying the Attention Weights Pruning problem, which is an approximation problem to a non-linear function. We provide an algorithm for obtaining the near-optimal pruning mask based on Gradient Descent (GD) with a convergence guarantee.

**Theorem 1.3** (Main result, informal version of Theorem 4.1). *For any $\epsilon > 0$, our Algorithm 1 can converge to the near-optimal pruning mask for the Attention Weights Pruning problem (Definition 1.2) in $O(d \operatorname{poly}(n)/\epsilon)$ time with $O(\xi + \epsilon)$ error, where $\xi$ is a small term depending on intrinsic property of the data and weights.*

In the above theorem, $\xi$ can be arbitrarily small as $\xi \to 0$ when the regularization coefficient $\lambda \to 0$. Thus, our analysis shows that although the objective function is highly non-linear, the GD training can converge to a near-optimal pruning mask solution.

Our experiments on synthetic clearly align and support our theoretical analysis (Section 6.1). Furthermore, we evaluate our non-linear pruning method and show it beyond SparseGPT and Wanda by a large margin on real data (C4 Dataset Raffel et al. (2020)) under real LLM (Llama 3.2-1B Meta (2024)) in Section 6.2 and Section 6.3.

Our contributions are as follows:

- This is the first work that analyzes the weights pruning problem based on Softmax attention, which is a non-linear function.

- We provide the closed form of the gradient of Attention Weights Pruning loss function (Theorem 5.3), and Lipschitz of that gradient (Theorem 5.4),

- We provide Gradient Descent based Algorithm 1 to obtain the near-optimal pruning mask and its convergence guarantee (Theorem 4.1).

- We conduct experiments to verify the effectiveness of our method (Section 6), showing our non-linear pruning method is beyond SparseGPT and Wanda by a large margin.

**Roadmap.** In Section 2, we review the related work. Section 3 introduces key concepts and definitions essential for the subsequent sections. In Section 4, we present our main result. Section 5 offers a technical overview of the methods employed. Experimental results are discussed in Section 6. Finally, Section 7 summarizes our findings and offers concluding remarks.

## 2 RELATED WORK

### 2.1 PRUNING AND COMPRESSION FOR LLMS

Model compression plays a critical role in improving the efficiency and deployment of large language models (LLMs) (Zhu et al., 2023) for its effectiveness in reducing computational overhead while preserving performance. Common compression techniques include quantization (Park et al., 2024; Xiao et al., 2023; Hooper et al., 2024), pruning (Chen et al., 2021; Hoefler et al., 2021; Hubara et al., 2021; Jin et al., 2022; Frantar & Alistarh, 2022; 2023; Sun et al., 2024; Li et al., 2024a; Zandieh et al., 2024; Zhang et al., 2024c; Xia et al., 2023; Ashkboos et al., 2024; Chen et al., 2025b), and knowledge distillation (Hsieh et al., 2023; Shridhar et al., 2023; Jiang et al., 2023; Wang et al., 2023). Specifically, pruning techniques have been developed extensively, such as unstructured pruning, which removes individual weights (Li et al., 2024a; Sun et al., 2024), and structured pruning, which eliminates entire components like neurons or attention heads (Michel et al., 2019; Ashkboos et al., 2024; Xia et al., 2024). Sun et al. (2024) proposed Wanda, a novel unstructured pruning technique that uses weight-activation products to induce up to 50% sparsity in LLMs without retraining, achieving competitive results with significantly lower computational cost. SparseGPT (Frantar & Alistarh, 2023) introduced a one-shot pruning method that achieves up to 60% sparsity in large GPT-family models with minimal impact on performance. A follow-up work (Li et al., 2024a) improved the complexity analysis of SparseGPT, reducing the running time from $O(d^3)$ to $O(d^{2.53})$, enabling faster pruning on LLMs. Together, these techniques contribute to more scalable and resource-efficient LLMs, maintaining competitive performance while substantially reducing computational resources.

## 2.2 Attention Acceleration

The attention mechanism has faced criticism due to its quadratic time complexity with respect to context length (Vaswani et al., 2017). Addressing this criticism, a variety of approaches are employed, including sparse attention (Hubara et al., 2021; Kurtic et al., 2023; Frantar & Alistarh, 2023; Li et al., 2024a), low-rank approximations (Razenshteyn et al., 2016; Li et al., 2016; Hu et al., 2022; Zeng & Lee, 2024; Hu et al., 2024d; Li et al., 2025a), and kernel-based methods (Charikar et al., 2020; Liu & Zenke, 2020; Deng et al., 2025; Zandieh et al., 2023; Liang et al., 2024b), to reduce computational overhead and improve scalability. Aggarwal & Alman (2022) enable the derivation of a low-rank representation of the attention matrix, which accelerates both the training and inference processes of single attention layer, tensor attention, and multi-layer transformer, achieving almost linear time complexity (Alman & Song, 2023; 2024a;b; Liang et al., 2024g;c; Ke et al., 2025a; Hu et al., 2024h). Other approaches like Mamba (Gu & Dao, 2023; Dao & Gu, 2024), Linearizing Transformers (Zhang et al., 2024b; Mercat et al., 2024), Hopfield Models (Hu et al., 2023; Wu et al., 2024b; Hu et al., 2024c; Xu et al., 2024a; Wu et al., 2024a; Hu et al., 2024a;b;f), and PolySketch-Former (Kacham et al., 2023) focus on architectural modifications and implementation optimizations to enhance performance. System-level optimizations such as FlashAttention (Dao et al., 2022; Dao, 2023; Shah et al., 2024) and block-wise parallel decoding (Stern et al., 2018) further improve efficiency. Collectively, these innovations have significantly augmented transformer models' ability to handle longer input sequences, unlocking broader applications across multiple sectors (Chen et al., 2023; Peng et al., 2023; Ding et al., 2024; Ma et al., 2024; Xu et al., 2024d; An et al., 2024; Jin et al., 2024; Li et al., 2024c; Liang et al., 2024d; Shi et al., 2024a).

## 3 Preliminary

**Notations.** For any positive integer $n$, we use $[n]$ to denote set $\{1, 2, \cdots, n\}$. For each $a, b \in \mathbb{R}^n$, we use $a \circ b \in \mathbb{R}^n$ to denote the Hadamard product, i.e., the $i$-th entry of $(a \circ b)$ is $a_i b_i$ for all $i \in [n]$. For $A \in \mathbb{R}^{m \times n}$, let $A_i \in \mathbb{R}^n$ denote the $i$-th row and $A_{*,j} \in \mathbb{R}^m$ denote the $j$-th column of $A$, where $i \in [m]$ and $j \in [n]$. We use $\exp(A)$ to denote a matrix where $\exp(A)_{i,j} := \exp(A_{i,j})$ for a matrix $A \in \mathbb{R}^{n \times d}$. We use $\|A\|_F$ to denote the Frobenius norm of a matrix $A \in \mathbb{R}^{n \times d}$, i.e., $\|A\|_F := \sqrt{\sum_{i \in [n]} \sum_{j \in [d]} |A_{i,j}|^2}$. For a symmetric matrix $A \in \mathbb{R}^{n \times n}$, $A \succeq 0$ means that $A$ is positive semidefinite (PSD), i.e., for all $x \in \mathbb{R}^n$, we have $x^\top A x \geq 0$.

**Attention weights pruning.** We aim to determine a near-optimal pruning mask $M$ for the attention weights in a self-attention mechanism. Furthermore, we incorporate causal attention masking[1] into our method to be more aligned with the current decoder-only LLM architecture, while our analysis can be applied to any general attention mask, e.g., block-wise attention mask. To formalize this, we provide the formal definition of causal attention mask and attention weights pruning in this section.

The causal attention mask ensures that each token in the sequence can attend only to itself and the preceding tokens. Here, we provide the formal definition of the causal attention mask:

**Definition 3.1** (Causal attention mask, Liang et al. (2024c)). *We define the causal attention mask as* $M_c \in \{0, 1\}^{n \times n}$, *where* $(M_c)_{i,j} = 1$ *if* $i \geq j$ *and* $(M_c)_{i,j} = 0$ *otherwise.*

Now, we incorporate Attention Weights Pruning (see Definition 1.2) with causal attention mask $M_c$.

**Definition 3.2** (Attention Weights Pruning with Causal Attention Mask). *Let* $M_c \in \{0, 1\}^{n \times n}$ *be the causal attention mask defined in Definition 3.1. Let* $A := \exp(XWX^\top) \circ M_c$ *and* $\widetilde{A} := \exp(X(M \circ W)X^\top) \circ M_c$. *Let* $D := \mathrm{diag}(A \cdot \mathbf{1}_n)$ *and* $\widetilde{D} := \mathrm{diag}(\widetilde{A} \cdot \mathbf{1}_n)$. *We define Attention Weights Pruning with Causal Attention Mask loss function to be* $\mathcal{L}(M) := \mathcal{L}_{\mathrm{attn}}(M) + \mathcal{L}_{\mathrm{reg}}(M)$ *where* $\mathcal{L}_{\mathrm{attn}}(M) := \frac{1}{2}\|D^{-1}A - \widetilde{D}^{-1}\widetilde{A}\|_F^2$ *and* $\mathcal{L}_{\mathrm{reg}}(M) := \frac{1}{2}\lambda\|M\|_F^2$.

---

[1] In this paper, we always use *pruning mask* to refer $M \in \mathbb{R}^{d \times d}$, which is our target, and use *causal attention mask* to refer $M_c \in \mathbb{R}^{n \times n}$, which is a fixed mask for standard self-attention.

**Algorithm 1** Gradient Descent for Pruning Mask (Theorem 4.1). Let $X_1, X_2, \ldots, X_k \in \mathbb{R}^{n \times d}$ be our calibration dataset of size $k$. We iteratively run our GD method on this dataset.

---

1: **procedure** PRUNEMASKGD( $X_1, X_2, \ldots, X_k \in \mathbb{R}^{n \times d}$, $W \in \mathbb{R}^{d \times d}$, $M_c \in \{0, 1\}^{d \times d}$, $\rho \in$
   $[0, 1], \lambda \in [0, 1], \epsilon \in (0, 0.1))$ ▷ Theorem 4.1
2:     Initialize $M \in \{1\}^{d \times d}$
3:     Initialize $\eta, T$ by Theorem 4.1
4:     **for** $j = 1 \to k$ **do** ▷ Iterate over the dataset
5:         $u_j \leftarrow \exp(X_j W X_j^\top)$
6:         $f_j \leftarrow \text{diag}((u_j \circ M_c) \cdot \mathbf{1}_n)^{-1}(u_j \circ M_c)$
7:     **end for**
8:     **for** $i = 1 \to T$ **do** ▷ Iterate over $T$ steps
9:         **for** $j = 1 \to k$ **do** ▷ Iterate over the dataset
10:           $\widetilde{u}_j \leftarrow \exp(X_j(M \circ W)X_j^\top)$
11:           $\widetilde{f}_j \leftarrow \text{diag}((\widetilde{u}_j \circ M_c) \cdot \mathbf{1}_n)^{-1}(\widetilde{u}_j \circ M_c)$
12:           $c_j \leftarrow \widetilde{f}_j - f_j$
13:           $p_{1,j} \leftarrow c_j \circ \widetilde{f}_j$
14:           $p_{2,j} \leftarrow \text{diag}(p_{1,j} \cdot \mathbf{1}_n)\widetilde{f}_j$
15:           $p_j \leftarrow p_{1,j} - p_{2,j}$
16:         **end for**
17:         $M \leftarrow M - (\eta/k) \cdot (W \circ (\sum_{j=1}^k X_j^\top p_j X_j) + \lambda M)$ ▷ Gradient Descent
18:     **end for**
19:     $m \leftarrow \text{vec}(M)$ ▷ Flatten $M$ into a vector
20:     $m_{\text{sorted}} \leftarrow \text{sort}(m)$ ▷ Sort the elements of $M$
21:     $\tau \leftarrow m_{\text{sorted}}[\lfloor \rho \cdot d^2 \rfloor]$ ▷ Get the $\rho$-th largest element
22:     $M_{ij} \leftarrow \begin{cases} 1 & \text{if } M_{ij} > \tau \\ 0 & \text{otherwise} \end{cases}$ ▷ Set the top $\rho$ entries to 1, others to 0
23:     **return** $M$
24: **end procedure**

---

## 4 MAIN RESULTS

We provide an Algorithm 1 for Attention Weights Pruning problem based on Gradient Descent (GD). We also prove the convergence for our GD algorithm in Theorem 4.1.

**Theorem 4.1** (Main result, formal version of Theorem 1.3)**.** *Let $M, X, W, \widetilde{D}, \widetilde{A}$ $\lambda$, $\mathcal{L}$, $\mathcal{L}_{\text{attn}}$ be defined in Definition 3.2. Assume $XX^\top \succeq \beta I$ and $\min_{i,j \in [n]}(\widetilde{D}^{-1}\widetilde{A})_{i,j} \geq \delta > 0$. Furthermore, Let $\mu = 2\min_{i,j \in [d]}\{|W_{i,j}|\} \cdot \beta \cdot \delta$. Let $\xi = 12n^{-1.5}\max_{i,j \in [d]}\{|W_{i,j}|\} \cdot \|X\|_F^2 \cdot \lambda d/\mu$. Then, for any $\epsilon > 0$, provided $\eta < 1/L$ where $L$ is the Lipschitz constant for $\nabla_M \mathcal{L}(M)$ (see Theorem 5.4), GD (Algorithm 1) with fixed step size $\eta$ and run for $T = 4\mathcal{L}(M^{(0)})/(\eta\mu\epsilon n^2)$ iterations results in the following guarantee,*

$$\frac{1}{n^2}\min_{t < T}\mathcal{L}_{\text{attn}}(M^{(t)}) + \frac{\lambda^2}{\mu n^2}\|M^{(t)}\|_F^2 \leq (\xi + \epsilon)/2.$$

*Proof.* Let $g(M) = 2\mathcal{L}_{\text{attn}}(M) + \frac{2\lambda^2}{\mu}\|M\|_F^2$. Note that $\mathcal{L}(M)$ satisfies the $(g(M), n^2\xi, 2, \mu)$-proxy PL inequality (Lemma 5.5). Also, we have $\mathcal{L}(M)$ is non-negative and has $L$-Lipschitz gradients Theorem 5.4. Thus, we finish the proof using Theorem 5.2. □

**Remark 4.2.** *The two assumptions in Theorem 4.1 are practical. The first assumption of the positive definite matrix is widely used in theoretical deep learning analysis, e.g., Li & Liang (2018); Du et al. (2019); Allen-Zhu et al. (2019b); Arora et al. (2019). The second assumption is natural, as $\widetilde{D}^{-1}\widetilde{A}$ is the pruned attention matrix, where each entry is*

$$\frac{\exp(X(M \circ W)X^\top)_{i,j}}{\sum_{j=1}^n \exp(X(M \circ W)X^\top)_{i,j}} > 0,$$

*which has a natural lower bound.*

Our error upper bound in Theorem 4.1 is $O(\xi + \epsilon)$, where $\epsilon$ can be arbitrarily small. For $\xi$, we can let it be small by choosing a proper $\lambda$, i.e., the $\xi$ error term can be made arbitrarily small by choosing small $\lambda$. However, if we choose a very small $\lambda$, the algorithm's run time gets larger as $T \propto 1/\eta \propto L \propto (\lambda + \text{other terms})$. Thus, although the objective function is highly non-linear, we can show that the Gradient Descent of our Algorithm 1 can converge to a good solution of the Attention Weights Pruning problem.

After solving the optimization problem, we obtain a pruning mask with real-valued entries. In practice, however, this pruning mask must be converted into a binary form, specifically $M \in \{0,1\}^{d \times d}$. We define the pruning ratio $\rho$ as the percentage of weights to be pruned. We apply this ratio by setting the pruning mask entries to zero for weights that fall below the $\rho$-th percentile and to one for those above. This ensures that only the specified proportion of weights is pruned.

## 5 TECHNIQUE OVERVIEW

In Section 5.1, we introduce some useful tools from previous work. In Section 5.2, we derive the close form of the gradient of Attention Weights Pruning. In Section 5.3, we calculate the Lipschitz constant of that gradient. In Section 5.4, we prove the PL inequality for our loss function.

### 5.1 PREVIOUS TOOLS ON CONVERGENCE OF GD

To analyze the convergence behavior of GD for our optimization problem (Definition 3.2), we first introduce the concept of $g$-proxy, $\xi$-optimal Polyak–Łojasiewicz(PL) inequality (Polyak, 1963; Lojasiewicz, 1963; Karimi et al., 2016), under which GD will converge:

**Definition 5.1** ($g$-proxy, $\xi$-optimal PL inequality, Definition 1.2 in Frei & Gu (2021)). *We say that a function $f : \mathbb{R}^p \to \mathbb{R}$ satisfies a g-proxy, $\xi$-optimal Polyak–Łojasiewicz inequality with parameters $\alpha > 0$ and $\mu > 0$ (in short, $f$ satisfies the $(g, \xi, \alpha, \mu)$-PL inequality) if there exists a function $g : \mathbb{R}^p \to \mathbb{R}$ and scalars $\xi \in \mathbb{R}$, $\mu > 0$ such that for all $w \in \mathbb{R}^p$, $\|\nabla f(w)\|^\alpha \geq \frac{1}{2}\mu(g(w) - \xi)$.*

PL inequality is a powerful tool for studying non-convex optimization, and it has been used in recent studies on provable guarantees for neural networks trained by gradient descent (Li & Liang, 2018; Allen-Zhu et al., 2019a;b;c; Frei et al., 2019; Cao & Gu, 2020; Ji & Telgarsky, 2019; Frei et al., 2021; Shi et al., 2021; 2024b). It provides a proxy convexity property, although the objective function is non-convex. In detail, for a function with good smoothness property, we can find some proxy functions and show the convergence by utilizing these proxy functions.

Leveraging this PL inequality, Frei & Gu (2021) derives the following GD convergence guarantees.

**Theorem 5.2** (Theorem 3.1 in Frei & Gu (2021)). *Suppose $f(w)$ satisfies the $(g(\cdot), \xi, \alpha, \mu)$-proxy PL inequality for some function $g(\cdot) : \mathbb{R}^p \to \mathbb{R}$. Assume that $f$ is non-negative and has L-Lipschitz gradients. Then for any $\epsilon > 0$, provided $\eta < 1/L$, GD with fixed step size $\eta$ and run for $T = 2\eta^{-1}(\mu\epsilon/2)^{-2/\alpha} f(w^{(0)})$ iterations results in the following guarantee, $\min_{t<T} g(w^{(t)}) \leq \xi + \epsilon$.*

The above theorem establishes that under the $(g, \xi, \alpha, \mu)$-PL inequality and Lipschitz continuity of the gradient, GD converges to a point where the proxy function $g(w)$ is within $\epsilon$ of $\xi$. To apply this result to our specific problem, we need to verify these conditions for our loss function $\mathcal{L}(M)$.

### 5.2 CLOSED FORM OF GRADIENT

As a first step, we compute the close form of the gradient $\nabla_M \mathcal{L}(M)$. The pruning mask $M$ is inside a non-linear function Softmax, which complicates our calculation. We defer the proof to Section C.

**Theorem 5.3** (Closed form of gradient, informal version of Theorem D.5). *Let $\mathcal{L}(M)$ be defined in Definition 3.2. Let $p$ be defined in Definition D.1. Let $X \in \mathbb{R}^{n \times d}$, $M \in [0,1]^{d \times d}$, $W \in \mathbb{R}^{d \times d}$. Then, we have*

$$\frac{\mathrm{d}\mathcal{L}(M)}{\mathrm{d}M} = W \circ (X^\top p X) + \lambda M.$$

Based on Theorem 5.3, we calculate the gradient of the pruning mask from Line 10 to Line 15 in our Algorithm 1.

## 5.3 LIPSCHITZ OF GRADIENT

Having obtained the close form of gradient, we proceed to investigate its Lipschitz continuity. We aim to show that the gradient $\nabla_M \mathcal{L}(M)$ is Lipschitz continuous with respect to $M$.

**Theorem 5.4** (Lipschitz of the gradient, informal version of Theorem F.8). *Let $R$ be some fixed constant that satisfies $R > 1$. Let $X \in \mathbb{R}^{n \times d}, W \in \mathbb{R}^{d \times d}$. We have $\|X\|_F \leq R$ and $\|W\|_F \leq R$. Let $\mathcal{L}(M)$ be defined in Definition 3.2. For $M, \widetilde{M} \in \mathbb{R}^{d \times d}$, we have*

$$\|\nabla_M \mathcal{L}(M) - \nabla_M \mathcal{L}(\widetilde{M})\|_F \leq (\lambda + 30dn^{7/2}R^6) \cdot \|M - \widetilde{M}\|_F.$$

We defer the proof to Section F. Establishing the Lipschitz continuity of the gradient satisfies one of the necessary conditions for applying Theorem 5.2. The above theorem implicates that the gradient for $M$ is upper bounded, providing a way to choose step size.

## 5.4 PL INEQUALITY OF GRADIENT

Next, we need to verify that our loss function satisfies the PL inequality with appropriate parameters. To complete the verification of the conditions required for convergence, we demonstrate that $\mathcal{L}(M)$ satisfies the PL inequality. We show that $\nabla_M \mathcal{L}(M)$ satisfies the PL inequality in this lemma:

**Lemma 5.5** (PL inequality, informal version of Lemma G.10). *Let $M, X, W, \widetilde{D}, \widetilde{A}, \lambda, \mathcal{L}, \mathcal{L}_{\text{attn}}$ be defined in Definition 3.2. Assume that $XX^\top \succeq \beta I$ and $\min_{i,j \in [n]} (\widetilde{D}^{-1}\widetilde{A})_{i,j} \geq \delta > 0$. Also,*

- *Let $\mu = 2\min_{i,j \in [d]}\{|W_{i,j}|\} \cdot \beta \cdot \delta$.*

- *Let $\xi = 12\sqrt{n}\max_{i,j \in [d]}\{|W_{i,j}|\} \cdot \|X\|_F^2 \cdot \lambda d/\mu$.*

*We have $\|\nabla_M \mathcal{L}(M)\|_F^2 \geq \frac{1}{2}\mu(2\mathcal{L}_{\text{attn}}(M) + \frac{2\lambda^2}{\mu}\|M\|_F^2 - \xi)$.*

We defer the proof to Section G. By confirming that $\mathcal{L}(M)$ satisfies the PL inequality and that its gradient is Lipschitz continuous, we then apply Theorem 5.2 to conclude that GD will converge to a solution within our desired error tolerance, and further prove Theorem 4.1.

To prove the PL inequality, we also need the following two key Lemmas, which introduce our two assumptions in our Theorem 4.1, $XX^\top \succeq \beta I$ and $\min_{i,j \in [n]}(\widetilde{D}^{-1}\widetilde{A})_{i,j} \geq \delta > 0$.

**Lemma 5.6** (Informal version of Lemma G.4). *Let $B \in \mathbb{R}^{n \times n}$ and $X \in \mathbb{R}^{n \times d}$. Assume that $XX^\top \succeq \beta I$. Then, we have $\|X^\top BX\|_F \geq \beta\|B\|_F$.*

**Lemma 5.7** (Informal version of Lemma G.7). *Let $B \in \mathbb{R}^{n \times n}$ and each row summation is zero, i.e., $B \cdot \mathbf{1}_n = \mathbf{0}_n$. Let $\widetilde{B} \in [0,1]^{n \times n}$ and each row summation is 1, i.e., $\widetilde{B} \cdot \mathbf{1}_n = \mathbf{1}_n$. Assume that $\min_{i,j \in [n]} \widetilde{B}_{i,j} \geq \delta > 0$. Then, we can show $\|B \circ \widetilde{B} - \text{diag}((B \circ \widetilde{B}) \cdot \mathbf{1}_n)\widetilde{B}\|_F \geq \delta \cdot \|B\|_F$.*

## 6 EXPERIMENT

### 6.1 EVALUATION ON SYNTHETIC DATA

We discuss the synthetic experiments conducted to illustrate the effectiveness of our Algorithm 1.

**Method and evaluation.** We implement our method following the pseudocode in Algorithm 1, using NumPy and JAX for acceleration. We evaluate our method on unstructured sparsity, meaning that zeros can occur anywhere within the attention weight matrix $W$. Specifically, we use Definition 3.2 as our loss function, optimizing over the pruning mask $M$ using gradient descent based on the closed-form expression derived in Theorem 5.3. To accelerate convergence, we leverage momentum into the optimization process and fix the momentum parameter at 0.9. After obtaining the optimal pruning mask, we convert $M$ to a binary pruning mask to prune $W$, maintaining sparsity at

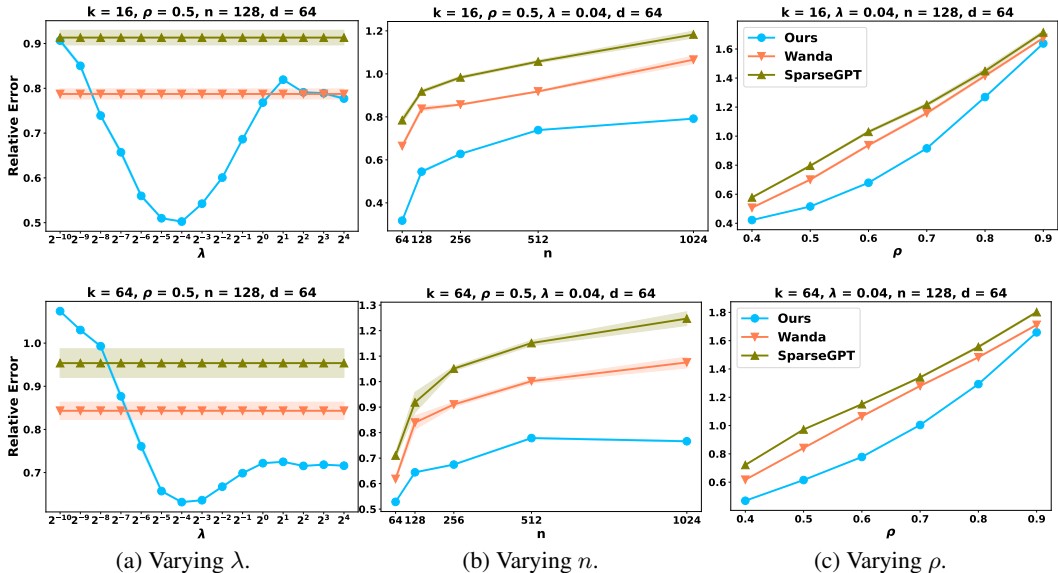

(a) Varying $\lambda$.    (b) Varying $n$.    (c) Varying $\rho$.

Figure 2: The comparison among our Algorithm 1, Wanda, and SparseGPT. The $y$-axis is a relative error, which is defined as $\frac{\|\widetilde{D}^{-1}\widetilde{A} - D^{-1}A\|_F^2}{\|D^{-1}A\|_F^2}$, where $D^{-1}A$ is original attention matrix and $\widetilde{D}^{-1}\widetilde{A}$ is approximated attention matrix based on three methods. We always use $d = 64$. We use $k = 16$ for the first row and $k = 64$ for the second row. The $x$-axis is (a) regularization coefficient $\lambda$ for the left column; (b) input sequence length $n$ for the middle column; (c) pruning ratio $\rho$ for the right column.

the desired pruning ratio $\rho$. We use the relative error as our evaluation metric, which is defined as $\|\widetilde{D}^{-1}\widetilde{A} - D^{-1}A\|_F^2 / \|D^{-1}A\|_F^2$, where $\widetilde{D}, \widetilde{A}, D, A$ are defined in Definition 3.2.

**Baselines.** We compare our method with two linear pruning approaches, namely Wanda (Sun et al., 2024) and SparseGPT (Frantar & Alistarh, 2023). Wanda is a pruning method that removes weights with the smallest magnitudes multiplied by the corresponding input activations, achieving sparsity without requiring retraining or weight updates. SparseGPT is a second-order pruning method that utilizes the Hessian matrix to prune a portion of the weight matrix while simultaneously updating the remaining parameters. We implement Wanda and SparseGPT as described in their respective papers. Notably, since the settings of SparseGPT and Wanda are linear, we do not prune the fused weight matrix $W$ directly; instead, we prune $W_Q$ and $W_K$ separately (see Figure 1).

**Data.** In order to assess the efficacy of different methods in approximating the attention matrix, we construct the data via a carefully defined generating process. Specifically, we create multiple independent random Gaussian matrices $G \in \mathbb{R}^{d \times d}$, where each entry of $G$ drawn from a normal distribution, i.e., $G_{i,j} \sim \mathcal{N}(0,1)$ for $i, j \in [d]$. Then, we perform singular value decomposition (SVD) on matrix $G$, i.e., $U, S, V^\top = \text{SVD}(G)$. We retain the first four singular values in $S$ and set others to zero, constraining the rank to four. Our $W_Q$ and $W_K$ are then constructed as $U \text{diag}(S) V^\top$. The weight matrix $W$ used in our setting is formed by taking the product $W = W_Q W_K^\top$. For $X \in \mathbb{R}^{n \times d}$, we generate it as a full-rank Gaussian random matrix.

**Setup.** In our experiments, the weight matrix dimension $d = 64$ is kept constant across all figures, and we simulate two datasets of size $k = 16$ and $k = 64$. We set the input sequence length $n = 128$ for experiments (a) and (c) in Figure 2. The pruning ratio $\rho = 0.5$ is set for experiments (a) and (b) in Figure 2. For our method, the regularization coefficient $\lambda := \widetilde{\lambda}/n$ where we abuse the notation to denote $\widetilde{\lambda}$ as the same used in Definition 3.2 and $\lambda$ here is the parameter we really control in experiments. $\lambda$ is set as $0.04$ for experiments (b) and (c) in Figure 2 (intuition drawn from experiment (a)). The total number of epochs is set as $T = 100$. The step size is set as $\eta = 0.1/\lambda$ because Theorem 4.1 indicates that $\eta$ is inversely proportional to $\lambda$ with some constant, i.e., $\eta \propto 1/L \propto 1/(\lambda + \text{other terms})$.

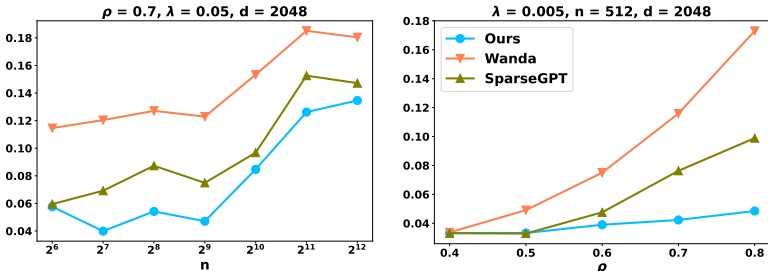

Figure 3: The comparison among our algorithm, Wanda, and SparseGPT on Llama 3.2-1B.

**Results.** Overall, the results in Figure 2 show that our Algorithm 1 outperforms Wanda and SparseGPT with a large margin, which supports our theoretical analysis in Theorem 4.1. In the following, we will discuss each setting in detail.

*Relation with regularization coefficient $\lambda$.* The leftmost column of Figure 2 investigates the impact of the regularization coefficient $\lambda$ on relative error. As $\lambda$ increases from very small values, the relative error initially decreases sharply for our algorithm, reaching a minimum before gradually rising again, which forms a $U$ shape curve. This behavior indicates that there is an optimal $\lambda$ where our algorithm achieves its best performance around $2^{-4}$. The U-shape curve phenomena are well-known in most hyper-parameter choosing, e.g., regularization coefficient.

*Relation with input sequence length $n$.* The center column of Figure 2 explores how the relative error changes with respect to the input sequence length $n$. As $n$ increases, the relative error for all three methods grows, though at different rates. Our method demonstrates a slower increase, maintaining a significant margin over both Wanda and SparseGPT, particularly for larger values of $n$. Wanda, while showing better performance than SparseGPT for larger sequence lengths, becomes comparable to SparseGPT as $n$ is relatively small.

*Relation with pruning ratio $\rho$.* The rightmost column of Figure 2 illustrates the relationship between the relative error and the pruning ratio $\rho$ for the three methods under comparison: our algorithm, Wanda, and SparseGPT. As the pruning ratio $\rho$ increases, all methods exhibit a rise in relative error, indicating a degradation in approximation accuracy. However, our algorithm consistently outperforms both Wanda and SparseGPT across the range of $\rho$, with a lower relative error. SparseGPT and Wanda follow a similar trend, closely tracking each other.

## 6.2 Experiment on Real Dataset and LLMs

In this subsection, we discuss the experiments conducted on the real dataset and LLMs to illustrate the effectiveness of our method.

**Method and evaluation.** We evaluate our method on unstructured sparsity, meaning that zeros can occur anywhere within the attention weight matrices $W_Q$ and $W_K$. Specifically, we use the loss function defined below:

$$\mathcal{L}(M_Q, M_K) := \frac{1}{2}\|D^{-1}A - \widetilde{D}^{-1}\widetilde{A}\|_F^2 + \frac{1}{2}(\|M_Q\|_F^2 + \|M_K\|_F^2) \tag{1}$$

where $D$ and $A$ are the original attention matrix, $M_Q, M_K$ are pruning masks, and

$$\widetilde{A} := \exp(X(M_Q \circ W_Q)(M_K \circ W_K)^\top X^\top), \quad \widetilde{D} := \mathrm{diag}(\widetilde{A} \cdot \mathbf{1}_n).$$

After obtaining the optimal pruning mask, we convert the pruning mask to a binary pruning mask to prune $W_Q$ and $W_K$, maintaining sparsity at the desired pruning ratio $\rho$. We use the relative error as our evaluation metric.

**Baselines.** The pruning is performed on the last attention layer of the pretrained model Llama 3.2-1B Meta (2024). The baselines are Wanda and SparseGPT, the same as Section 6.2.

**Data.** To simulate real-world large language models, we utilize the Colossal Clean Crawled Corpus (C4 Dataset) Raffel et al. (2020), which is also used as the calibration dataset in our baselines Wanda and SparseGPT. Additionally, with a primary focus on pruning the attention matrix, we extract the

input hidden states corresponding to the target attention matrix from the pretrained Llama 3.2 model using a customized hook function and use these as our input $X$.

**Setup.** The weight matrix dimension is $d = 2048$ in Llama 3.2-1B. For all the experiments, we set $\lambda$ as $0.05$ and $\eta$ as $0.005$. For the varying $n$ experiment, we set the pruning ratio $\rho$ as $0.7$. For the varying $\rho$ experiment, we set the input sequence length $n$ as $512$.

**Results.** Overall, the results in Figure 3 show that our algorithm outperforms Wanda and SparseGPT in real-world LLMs, which supports our theoretical analysis in Theorem 4.1 and enhances our preliminary experiment in Section 6. In the following, we will discuss each setting in detail.

*Relation with input sequence length $n$.* The left column of Figure 3 shows that our algorithm outperforms the baselines in different sequence lengths continuously.

*Relation with pruning ratio $\rho$.* The right column of Figure 3 illustrates that as the pruning ratio $\rho$ increases, all methods exhibit a rise in relative error. But our algorithm consistently outperforms both Wanda and SparseGPT across the range of $\rho$ with a much lower increasing rate.

*Assumptions verification.* Notice that Theorem 4.1 relies on two assumptions: $XX^\top \succeq \beta I$ and $\min_{i,j \in [n]}(\widetilde{D}^{-1}\widetilde{A})_{i,j} \geq \delta > 0$. We verify these assumptions using the C4 dataset, obtaining $\beta \approx 0.034$ and $\delta \approx 0.0025$, thereby demonstrating the practicality of our theoretical framework.

### 6.3 EXPERIMENT ON END-TO-END PERPLEXITY

We present the experiment on end-to-end perplexity in this section.

**Baselines.** As our method focuses on pruning the attention matrix, it can be seamlessly combined with approaches that perform linear pruning on the MLP, such as Wanda and SparseGPT. Therefore, we conduct three groups of experiments: (1) using a dense MLP without pruning, (2) applying SparseGPT to prune the MLP layer, and (3) using Wanda to prune the MLP layer. Subsequently, we apply Wanda, SparseGPT, and our method to prune the attention weights.

**Method and data.** We use the same method and data as Section 6.2.

Table 1: Comparison of different methods based on Perplexity (PPL).

| Method | PPL |
|---|---|
| Dense MLP + Dense Attn | 12.487 |
| Dense MLP + SparseGPT Attn | 14.269 |
| Dense MLP + Wanda Attn | 14.912 |
| Dense MLP + Our Attn | **13.885** |
| Wanda MLP + Wanda Attn | 30.426 |
| Wanda MLP + SparseGPT Attn | 26.074 |
| Wanda MLP + Our Attn | **24.427** |
| SparseGPT MLP + Wanda Attn | 45.435 |
| SparseGPT MLP + SparseGPT Attn | 36.641 |
| SparseGPT MLP + Our Attn | **34.946** |

**Setup.** We use Llama 3.2 1B as the target model to prune, in which the hidden state dimension is 2048. We set the pruning ratio as $0.5$ for MLP and $0.5$ for Attention Layer, when pruning MLP and attention weights at the same time, we have pruning ratio $\rho$ of the whole model as $0.5$. We use $8$ sentences in the C4 training dataset Raffel et al. (2020) as calibration data, and we use $128$ sentences in the C4 validation dataset Raffel et al. (2020) to evaluate perplexity. We set $0.005$ as the learning rate $\eta$ and $0.05$ as the regularization coefficient $\lambda$.

**Results.** As we can see in Table 1, our Attention pruning methods always outperform Wanda Attention pruning and SparseGPT Attention pruning by a large margin when combined with Dense/Wanda/SparseGPT MLP pruning methods. These empirical results support our theoretical analysis that our pruning method can converge. Furthermore, our method is broadly applicable in the real-world case and can be combined with many other pruning methods.

## 7 CONCLUSION

This paper introduces a novel approach to LLM weight pruning that directly optimizes for approximating the attention matrix. We provide theoretical guarantees for the convergence of our Gradient Descent-based algorithm to a near-optimal pruning mask solution. Experimental results demonstrated the method's effectiveness in maintaining model performance while reducing computational costs. This work establishes a new theoretical foundation for pruning algorithm design in LLMs, potentially enabling more efficient inference on resource-constrained devices.

ACKNOWLEDGEMENT

Research is partially supported by the National Science Foundation (NSF) Grants 2023239-DMS, CCF-2046710, and Air Force Grant FA9550-18-1-0166.

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

# Appendix

## CONTENTS

**Roadmap.** The appendix is organized as follows. In Section A, we reviewed more literature related to our paper. In Section B, we give the preliminary of our paper. In Section C, we provide a detailed gradient analysis of the loss function. In Section D, we provide details about how we integrate the gradient of loss function into matrix form. In Section E, we bound some basic functions to be used later. In Section F, we provide proof for the Lipschitz property of the gradient of the loss function. In Section G, we provide proof of convergence for GD.

## A  MORE RELATED WORK

**Large Language Models.** Transformer-based neural networks (Vaswani et al., 2017) have rapidly emerged as the dominant architecture for natural language processing in machine learning. When expanded to billions of parameters and trained on vast, diverse datasets, these systems are typically termed large language models (LLMs) or foundation models (Bommasani et al., 2021). Prominent LLM examples encompass BERT (Devlin et al., 2019), PaLM (Chowdhery et al., 2022), Llama (Touvron et al., 2023; Meta, 2024), and GPT4o (OpenAI, 2024a), which display adaptable competencies (Bubeck et al., 2023) across numerous downstream applications. To enhance LLMs for domain-specific uses, researchers have created multiple adaptation approaches. These include: adapter modules (Hu et al., 2022; Gao et al., 2023a; Zhang et al., 2023b; Shi et al., 2023); calibration mechanisms (Zhou et al., 2023; Zhao et al., 2021); multitask refinement (Gao et al., 2021; Von Oswald et al., 2023; Xu et al., 2024d; 2023); along with prompt engineering (Lester et al., 2021), scratchpad approaches (Nye et al., 2021), instruction optimization (Chung et al., 2022; Li & Liang, 2021; Mishra et al., 2022), symbolic adaptation (Wei et al., 2023; Xu et al., 2024b; 2022), blackbox adjustments (Sun et al., 2022), human-aligned reinforcement learning (Ouyang et al., 2022), and structured reasoning techniques (Khattab et al., 2022; Yao et al., 2023; Wei et al., 2022; Zheng et al., 2024). Contemporary investigations cover tensor architecture innovations Alman & Song (2024b); Liang et al. (2024g); Sanford et al. (2024); Zhang et al. (2025), efficiency enhancements Alman & Song (2024c); Chen et al. (2024c; 2025b); Hu et al. (2024a;b;c;h; 2023); Ke et al. (2024); Liang et al. (2024a); Li et al. (2024b;c;d;a); Liang et al. (2024d); Shi et al. (2024a); Shen et al. (2024a;b); Song et al. (2024); Wu et al. (2024a;b); Xu et al. (2024a), plus ancillary studies Chen et al. (2024a); Demirel et al. (2022); Chang et al. (2024); Deng et al. (2022); Gao et al. (2023b;d); Li et al. (2025b;d); Liang et al. (2024f); Li et al. (2024e); Shrivastava et al. (2023); Sinha et al. (2023); Song & Yang (2023); Tan et al. (2023); Xie et al. (2022); Xu et al. (2024c); Zhang (2024); Zhang et al. (2024a); Chen et al. (2025d); Li et al. (2025c); Ke et al. (2025b); Liang et al. (2025); Chen et al. (2025a;c); Gao et al. (2023c); Hu et al. (2024g;e); Wu et al. (2024c).

## B  PRELIMINARY

In Section B.1, we introduce some notations we use in this paper. In Section B.2, we provide some basic facts.

### B.1  NOTATIONS

For any positive integer $n$, we use $[n]$ to denote set $\{1, 2, \cdots, n\}$. For two vectors $x \in \mathbb{R}^n$ and $y \in \mathbb{R}^n$, we use $\langle x, y \rangle$ to denote the inner product between $x, y$, i.e., $\langle x, y \rangle = \sum_{i=1}^n x_i y_i$. For each $a, b \in \mathbb{R}^n$, we use $a \circ b \in \mathbb{R}^n$ to denote the Hadamard product, i.e. the $i$-th entry of $(a \circ b)$ is $a_i b_i$ for all $i \in [n]$. We use $e_i$ to denote a vector where only $i$-th coordinate is 1, and other entries are 0. We use $\mathbf{1}_n$ to denote a length-$n$ vector where all the entries are ones. We use $\|x\|_p$ to denote the $\ell_p$ norm of a vector $x \in \mathbb{R}^n$, i.e. $\|x\|_1 := \sum_{i=1}^n |x_i|$, $\|x\|_2 := (\sum_{i=1}^n x_i^2)^{1/2}$, and $\|x\|_\infty := \max_{i \in [n]} |x_i|$. For $A \in \mathbb{R}^{m \times n}$, let $A_i \in \mathbb{R}^n$ denote the $i$-th row and $A_{*,j} \in \mathbb{R}^m$ denote the $j$-th column of $A$, where $i \in [m]$ and $j \in [n]$. For a square matrix $A$, we use $\mathrm{tr}[A]$ to denote the trace of $A$, i.e., $\mathrm{tr}[A] = \sum_{i=1}^n A_{i,i}$. For two matrices $X, Y \in \mathbb{R}^{m \times n}$, the standard inner product between matrices is defined by $\langle X, Y \rangle := \mathrm{tr}[X^\top Y]$. We use $\exp(A)$ to denote a matrix where $\exp(A)_{i,j} := \exp(A_{i,j})$ for a matrix $A \in \mathbb{R}^{n \times d}$. For $k > n$, for any matrix $A \in \mathbb{R}^{k \times n}$, we use $\|A\|$ to denote the spectral norm of $A$, i.e. $\|A\| := \sup_{x \in \mathbb{R}^n} \|Ax\|_2 / \|x\|_2$. We use $\|A\|_\infty$ to denote the $\ell_\infty$ norm of a matrix $A \in \mathbb{R}^{n \times d}$, i.e. $\|A\|_\infty := \max_{i \in [n], j \in [d]} |A_{i,j}|$. We use $\|A\|_F$ to denote the Frobenius norm of a matrix $A \in \mathbb{R}^{n \times d}$, i.e. $\|A\|_F := \sqrt{\sum_{i \in [n]} \sum_{j \in [d]} |A_{i,j}|^2}$. For a symmetric matrix $A \in \mathbb{R}^{n \times n}$,

we use $A \succeq 0$ (positive semidefinite (PSD)), if for all $x \in \mathbb{R}^n$, we have $x^\top A x \geq 0$. We use $\lambda_{\min}(A)$ and $\lambda_{\max}(A)$ to denote the minimum and the maximum eigenvalue of the square matrix $A$, respectively. Let $A \in \mathbb{R}^{n \times d}$. We use $a := \text{vec}(A)$ to denote a length $nd$ vector. We stack rows of $A$ into a column vector, i.e. $\text{vec}(A) := [a_1^\top, a_2^\top, \ldots, a_n^\top]^\top$ where $a_i^\top$ is the $i$-th row of $A$, or simply $\text{vec}(A)_{j+(i-1)d} := A_{i,j}$ for any $i \in [n], j \in [d]$.

## B.2 FACTS

**Fact B.1** (Indexing). *Suppose we have matrices $U \in \mathbb{R}^{n \times m}, V \in \mathbb{R}^{m \times d}$. We define*

$$\underbrace{X}_{n \times d} := \underbrace{U}_{n \times m} \underbrace{V}_{m \times d}.$$

*Then, we have the following:*

- *Indexing for one row: $X_i = V^\top U_i \in \mathbb{R}^d$, i.e. $X_i^\top = U_i^\top V$, for $i \in [n]$.*

- *Indexing for one column: $X_{*,j} = U V_{*,j} \in \mathbb{R}^n$ for $j \in [d]$.*

**Fact B.2.** *We have*

**Part 1.** *Suppose we have vectors $u \in \mathbb{R}^n, v \in \mathbb{R}^n$. For $i \in [n]$, we define*

$$x_i := u_i v_i.$$

*Then we have the following:*

- $\underbrace{x}_{n \times 1} = \underbrace{u \circ v}_{n \times 1} = \underbrace{\text{diag}(u)}_{n \times n} \underbrace{v}_{n \times 1} = \underbrace{\text{diag}(v)}_{n \times n} \underbrace{u}_{n \times 1}$

**Part 2.** *Suppose we have matrix $W \in \mathbb{R}^{n \times n}$, vector $u \in \mathbb{R}^n$. For $i \in [n]$, we define*

$$X_{*,j} = W_{*,j} u_j.$$

*Then we have the following:*

- $X = W \text{diag}(u)$

**Fact B.3** (Calculus). *We have*

**Part 1.** (Scalar calculus) *For any $t \in \mathbb{R}$, function $f : \mathbb{R} \to \mathbb{R}$, we have*

- $\frac{\mathrm{d} f^n(t)}{\mathrm{d} t} = n f^{n-1}(t) \frac{\mathrm{d} f(t)}{\mathrm{d} t}$.

**Part 2.** (Vector calculus) *For any $x, y \in \mathbb{R}^n$, $t \in \mathbb{R}$, we have*

- $\frac{\mathrm{d}(x \circ y)}{\mathrm{d} t} = \frac{\mathrm{d} x}{\mathrm{d} t} \circ y + \frac{\mathrm{d} y}{\mathrm{d} t} \circ x$. *(Product rule of vector Hadamard product)*

- $\frac{\mathrm{d}\langle x, y \rangle}{\mathrm{d} t} = \langle \frac{\mathrm{d} x}{\mathrm{d} t}, y \rangle + \langle x, \frac{\mathrm{d} y}{\mathrm{d} t} \rangle$. *(Product rule of inner product)*

- $\frac{\mathrm{d} x}{\mathrm{d} x_i} = e_i$.

**Part 3.** (Matrix calculus) *For any $X, Y \in \mathbb{R}^{n \times m}$, $Z \in \mathbb{R}^{m \times d}$, $t \in \mathbb{R}$ which is independent of $Z$, function $f : \mathbb{R} \to \mathbb{R}^{n \times d}$, functions $f_1(t), f_2(t), \ldots, f_n(t) : \mathbb{R} \to \mathbb{R}^{n \times d}$, we have*

- $\frac{\mathrm{d}(X \circ Y)}{\mathrm{d} t} = \frac{\mathrm{d} X}{\mathrm{d} t} \circ Y + \frac{\mathrm{d} Y}{\mathrm{d} t} \circ X$. *(Product rule of matrix Hadamard product)*

- $\frac{\mathrm{d} \exp(f(t))}{\mathrm{d} t} = \exp(f(t)) \circ \frac{\mathrm{d} f(t)}{\mathrm{d} t}$, *where $\exp(\cdot)$ is applied entry-wise.*

- $\frac{\mathrm{d}(XZ)}{\mathrm{d} t} = \frac{\mathrm{d} X}{\mathrm{d} t} Z$.

- $\frac{\mathrm{d}(ZX^\top)}{\mathrm{d} t} = Z \frac{\mathrm{d} X^\top}{\mathrm{d} t}$.

- $\frac{\mathrm{d}}{\mathrm{d}t} \sum_{i=1}^{n} f_i(t) = \sum_{i=1}^{n} \frac{\mathrm{d}f_i(t)}{\mathrm{d}t}$.

- $\underbrace{\frac{\mathrm{d}X}{\mathrm{d}X_{i,j}}}_{n \times m} = \underbrace{e_i}_{n \times 1} \underbrace{e_j^\top}_{1 \times m}$.

**Fact B.4** (Basic algebra). *Let $u \in \mathbb{R}^n$, $v \in \mathbb{R}^n$, $w \in \mathbb{R}^n$, $X \in \mathbb{R}^{n \times d}$, $Y \in \mathbb{R}^{n \times d}$, and $Z \in \mathbb{R}^{n \times n}$. Then, we have*

- $\langle u, v \rangle = \langle v, u \rangle = u^\top v = v^\top u$

- $u \circ v = v \circ u = \mathrm{diag}(u)v = \mathrm{diag}(v)u$

- $\langle u, v \rangle = \langle u \circ v, \mathbf{1}_n \rangle$

- $\langle u \circ v, w \rangle = \langle u \circ w, v \rangle = \langle w \circ v, u \rangle$

- $u^\top (v \circ w) = u^\top \mathrm{diag}(v)w$

- $(X \circ Y)^\top = X^\top \circ Y^\top$

- $X \circ e_i e_j^\top = X_{i,j} e_i e_j^\top$

- $\mathrm{diag}(u)Z\,\mathrm{diag}(v) = (uv^\top) \circ Z$

- $XY^\top = \sum_{i \in [d]} X_{*,i} Y_{*,i}^\top$

- $X_{i,j} Y_{i,j} = (X \circ Y)_{i,j}$

- $\sum_{j \in [n]} u \circ A_{*,j} = u \circ \sum_{j \in [n]} A_{*,j}$

- $\|X\|_F^2 = \mathrm{tr}[XX^\top]$

- $\mathrm{tr}[XY^\top] = \mathrm{tr}[Y^\top X]$

- $\|\mathrm{diag}(u)\|_F = \|u\|_2$

**Fact B.5** (Norm bounds). *For $a \in \mathbb{R}$, $u \in \mathbb{R}^d$, $X, Y \in \mathbb{R}^{n \times d}$, $Z \in \mathbb{R}^{d \times m}$ we have*

- $\|aX\|_F = |a|\|X\|_F$ *(absolute homogeneity)*.

- $\|X + Y\|_F \leq \|X\|_F + \|Y\|_F$ *(triangle inequality)*.

- $|\langle X, Y \rangle| \leq \|X\|_F \cdot \|Y\|_F$ *(Cauchy–Schwarz inequality)*.

- $\|X^\top\|_F = \|X\|_F$.

- $\|Xu\|_2 \leq \|X\| \cdot \|u\|_2$

- $\|X \circ Y\|_F \leq \|X\|_F \cdot \|Y\|_F$.

- *For any $i \in [n]$, $j \in [d]$, we have $|X_{i,j}| \leq \|X\|_F$.*

- $\|X\| \leq \|X\|_F \leq \sqrt{k}\|X\|$ *where $k$ is the rank of $X$.*

- $\|Y \cdot Z\|_F \leq \|Y\|_F \cdot \|Z\|_F$.

**Fact B.6.** *For matrices $A, B \in \mathbb{R}^{m \times n}$, we have*

$$\|A + B\|_F^2 = \|A\|_F^2 + \|B\|_F^2 + 2\langle A, B \rangle.$$

*Proof.* We can show

$$\begin{aligned}
\|A + B\|_F^2 &= \mathrm{tr}[(A + B)^\top (A + B)] \\
&= \mathrm{tr}[A^\top A + A^\top B + B^\top A + B^\top B]
\end{aligned}$$

$$= \text{tr}[A^\top A] + \text{tr}[B^\top B] + 2\,\text{tr}[A^\top B]$$
$$= \|A\|_F^2 + \|B\|_F^2 + 2\,\text{tr}[A^\top B]$$
$$= \|A\|_F^2 + \|B\|_F^2 + 2\langle A, B\rangle$$

where the first step follows from $\text{tr}[A^\top A] = \|A\|_F^2$ for matrix $A \in \mathbb{R}^{m \times n}$, the second step follows from the basic algebra, the third follows from $\text{tr}[X^\top Y] = \text{tr}[XY^\top]$ for matrices $X, Y \in \mathbb{R}^{m \times n}$, the fourth step follows from $\text{tr}[A^\top A] = \|A\|_F^2$ for matrix $A \in \mathbb{R}^{m \times n}$, and the last step follows from definition of inner product of matrices. $\qquad\square$

**Lemma B.7.** *Let $M \in \mathbb{R}^{n \times n}$. Let $X \in \mathbb{R}^{n \times n}$ be independent of $M$. We have*

$$\frac{\mathrm{d}(M \circ X)}{\mathrm{d}M_{i,j}} = X_{i,j} e_i e_j^\top$$

*Proof.* We can show

$$\frac{\mathrm{d}(M \circ X)}{\mathrm{d}M_{i,j}} = M \circ \frac{\mathrm{d}X}{\mathrm{d}M_{i,j}} + X \circ \frac{\mathrm{d}M}{\mathrm{d}M_{i,j}}$$
$$= X \circ \frac{\mathrm{d}M}{\mathrm{d}M_{i,j}}$$
$$= X \circ (e_i e_j^\top)$$
$$= X_{i,j} e_i e_j^\top$$

where the first step, the second step, and the third step follow from Fact B.3, the fourth step follows from Fact B.4.

$$\square$$

## C  GRADIENT CALCULATION

### C.1  DEFINITIONS

In this section, we introduce some definitions we used to compute $\frac{\mathrm{d}\mathcal{L}(M)}{\mathrm{d}M}$. First, we introduce the exponential function.

**Definition C.1** (Exponential function $u, \widetilde{u}$). *If the following conditions hold*

- *Let $X \in \mathbb{R}^{n \times d}$.*

- *Let $W \in \mathbb{R}^{d \times d}$.*

- *Let $M \in [0,1]^{d \times d}$.*

- *Let $i_0 \in [n]$.*

*We define $u \in \mathbb{R}^{n \times n}$ as follows*

$$u := \exp(XWX^\top).$$

*We define $\widetilde{u}(M) \in \mathbb{R}^{n \times n}$ as follows*

$$\widetilde{u}(M) := \exp(X(M \circ W)X^\top).$$

*We define $i_0$-th row of $\widetilde{u}(M)$ as follows*

$$\widetilde{u}(M)_{i_0} := \exp(X(M \circ W)X^\top)_{i_0}.$$

Then, we introduce the sum function.

**Definition C.2** (Sum function of softmax $\alpha, \widetilde{\alpha}$). *If the following conditions hold*

- *Let $M \in [0,1]^{d \times d}$.*

- Let $M_c \in \{0,1\}^{n \times n}$ be the causal attention mask defined in Definition 3.1.

- Let $u, \widetilde{u}(M)$ be defined as Definition C.1.

- Let $i_0 \in [n]$.

*We define $\alpha \in \mathbb{R}^n$ as follows*

$$\alpha := (u \circ M_c) \cdot \mathbf{1}_n.$$

*We define $\widetilde{\alpha}(M) \in \mathbb{R}^n$ as follows*

$$\widetilde{\alpha}(M) := (\widetilde{u}(M) \circ M_c) \cdot \mathbf{1}_n.$$

*We define $i_0$-th entry of $\widetilde{\alpha}(M)$ as follows*

$$\widetilde{\alpha}(M)_{i_0} := \langle (\widetilde{u}(M) \circ M_c)_{i_0}, \mathbf{1}_n \rangle$$

Then, we introduce the Softmax probability function.

**Definition C.3** (Softmax probability function $f$, $\widetilde{f}$). *If the following conditions hold*

- Let $M \in [0,1]^{d \times d}$.

- Let $M_c \in \{0,1\}^{n \times n}$ be the causal attention mask defined in Definition 3.1.

- Let $u, \widetilde{u}(M)$ be defined as Definition C.1.

- Let $\alpha, \widetilde{\alpha}(M)$ be defined as Definition C.2.

- Let $i_0, j_0 \in [n]$.

*We define $f \in \mathbb{R}^{n \times n}$ for each $j \in [n]$ as follows*

$$f := \operatorname{diag}(\alpha)^{-1}(u \circ M_c).$$

*We define $\widetilde{f}(M) \in \mathbb{R}^{n \times n}$ for each $j \in [n]$ as follows*

$$\widetilde{f}(M) := \operatorname{diag}(\widetilde{\alpha}(M))^{-1}(\widetilde{u}(M) \circ M_c).$$

*We define $i_0$-th row of $\widetilde{f}(M)$ as follows*

$$\widetilde{f}(M)_{i_0} := \widetilde{\alpha}(M)_{i_0}^{-1}(\widetilde{u}(M) \circ M_c)_{i_0}.$$

*We define the entry in $i_0$-th row, $j_0$-th column of $\widetilde{f}(M)$ as follows*

$$\widetilde{f}(M)_{i_0,j_0} := \widetilde{\alpha}(M)_{i_0}^{-1}(\widetilde{u}(M) \circ M_c)_{i_0,j_0}.$$

Then, we introduce the one-unit loss function.

**Definition C.4** (One unit loss function $c$). *If the following conditions hold*

- Let $f$, $\widetilde{f}$ be defined in Definition C.3.

- Let $M \in [0,1]^{d \times d}$.

- Let $i_0, j_0 \in [n]$.

*We define $c(M) \in \mathbb{R}^{n \times n}$ as follows*

$$c(M) := \widetilde{f}(M) - f$$

*We define $i_0$-th row of $c(M)$ as follows*

$$c(M)_{i_0} := \widetilde{f}(M)_{i_0} - f_{i_0}$$

*We define $j_0$-th column of $c(M)$ as follows*

$$c(M)_{*,j_0} := \widetilde{f}(M)_{*,j_0} - f_{*,j_0}$$

*We define the entry in $i_0$-th row, $j_0$-th column of $c(M)$ as follows*

$$c(M)_{i_0,j_0} := \widetilde{f}(M)_{i_0,j_0} - f_{i_0,j_0}$$

Then, we introduce the reconstruction error.

**Definition C.5** (Reconstruction Error $\mathcal{L}_{\mathrm{attn}}$)**.** *If the following conditions hold*

- *Let $M \in [0,1]^{d \times d}$.*

- *Let $c(M)$ be defined in Definition C.4.*

*We define $\mathcal{L}_{\mathrm{attn}}(M) \in \mathbb{R}$ as follows*

$$\mathcal{L}_{\mathrm{attn}}(M) := \frac{1}{2} \|c(M)\|_F^2 = \frac{1}{2} \sum_{i_0=1}^{n} \sum_{j_0=1}^{n} c(M)_{i_0,j_0}^2.$$

Then, we introduce the regularization term.

**Definition C.6** (Regularization Term $\mathcal{L}_{\mathrm{reg}}$)**.** *If the following conditions hold*

- *$M \in [0,1]^{d \times d}$.*

*We define $\mathcal{L}_{\mathrm{reg}}(M) \in \mathbb{R}$ as follows*

$$\mathcal{L}_{\mathrm{reg}}(M) := \frac{1}{2} \lambda \|M\|_F^2.$$

Finally, we introduce the overall loss function.

**Definition C.7** (Overall loss function $\mathcal{L}$)**.** *If the following conditions hold*

- *Let $M \in [0,1]^{d \times d}$.*

- *Let $\mathcal{L}_{\mathrm{attn}}(M)$ be defined in Definition C.5.*

- *Let $\mathcal{L}_{\mathrm{reg}}(M)$ be defined in Definition C.6.*

- *Let $\lambda \in \mathbb{R}_+$ be the regularization parameter.*

*We define $\mathcal{L}(M)$ as follows*

$$\mathcal{L}(M) := \mathcal{L}_{\mathrm{attn}}(M) + \mathcal{L}_{\mathrm{reg}}(M)$$

## C.2 GRADIENT FOR EACH ROW OF $X(M \circ W)X^\top$

We introduce the Lemma of gradient for each row of $X(M \circ W)X^\top$.

**Lemma C.8.** *Let $i_1 \in [d]$, $j_1 \in [d]$, $i_0 \in [n]$, we have*

$$\underbrace{\frac{\mathrm{d}(X(M \circ W)X^\top)_{i_0}}{\mathrm{d}M_{i_1,j_1}}}_{n \times 1} = \underbrace{W_{i_1,j_1}}_{\text{scalar}} \underbrace{X_{i_0,i_1}}_{\text{scalar}} \underbrace{X_{*,j_1}}_{n \times 1}$$

*Proof.* We can simplify the derivative expression

$$
\begin{aligned}
\frac{\mathrm{d}(X(M \circ W)X^\top)_{i_0}}{\mathrm{d}M_{i_1,j_1}} &= \frac{\mathrm{d}X(X(M \circ W))_{i_0}}{\mathrm{d}M_{i_1,j_1}} \\
&= \frac{\mathrm{d}X(M \circ W)^\top X_{i_0}}{\mathrm{d}M_{i_1,j_1}} \\
&= X \frac{\mathrm{d}(M \circ W)^\top}{\mathrm{d}M_{i_1,j_1}} X_{i_0}
\end{aligned}
\tag{2}
$$

where the first and second step follows from Fact B.1, the third step follows from Fact B.3.

We further compute Eq. (2):

$$\frac{\mathrm{d}(M \circ W)^\top}{\mathrm{d}M_{i_1,j_1}} = \frac{\mathrm{d}M^\top \circ W^\top}{\mathrm{d}M_{i_1,j_1}}$$

$$
\begin{aligned}
&= \frac{\mathrm{d} M^\top \circ W^\top}{\mathrm{d}(M^\top)_{j_1,i_1}} \\
&= (W^\top)_{j_1,i_1} e_{j_1} e_{i_1}^\top \\
&= W_{i_1,j_1} e_{j_1} e_{i_1}^\top
\end{aligned}
\tag{3}
$$

where the first follows from Fact B.4, the second step follows from for any matrix $X$, $X_{i,j} = (X^\top)_{j,i}$, the third step follows from Fact B.7, and the fourth step follows from for any matrix $X$, $X_{i,j} = (X^\top)_{j,i}$.

Finally, we have

$$
\begin{aligned}
\frac{\mathrm{d}(X(M \circ W)X^\top)_{i_0}}{\mathrm{d} M_{i_1,j_1}} &= X W_{i_1,j_1} e_{j_1} e_{i_1}^\top X_{i_0} \\
&= W_{i_1,j_1} (X e_{j_1})(e_{i_1}^\top X_{i_0}) \\
&= W_{i_1,j_1} X_{*,j_1} X_{i_0,i_1}
\end{aligned}
$$

where the first step follows from Eq. (2) and Eq. (3), and the second step and the third step follow from basic algebra. $\qquad\square$

We introduce the Lemma of the gradient for each row of $\widetilde{u}(M)$.

### C.3   GRADIENT FOR EACH ROW OF $\widetilde{u}(M)$

**Lemma C.9.** *If the following conditions hold:*

- *Let $\widetilde{u}(M)$ be defined in Definition C.1.*

*Let $i_1 \in [d]$, $j_1 \in [d]$, $i_0 \in [n]$, we have*

$$
\underbrace{\frac{\mathrm{d}\widetilde{u}(M)_{i_0}}{\mathrm{d} M_{i_1,j_1}}}_{n \times 1} = \underbrace{\widetilde{u}(M)_{i_0}}_{n \times 1} \circ (\underbrace{W_{i_1,j_1}}_{\text{scalar}} \underbrace{X_{i_0,i_1}}_{\text{scalar}} \underbrace{X_{*,j_1}}_{n \times 1})
$$

*Proof.* We have

$$
\begin{aligned}
\underbrace{\frac{\mathrm{d}\widetilde{u}(M)_{i_0}}{\mathrm{d} M_{i_1,j_1}}}_{n \times 1} &= \underbrace{\frac{\mathrm{d}\exp(X(M \circ W)X^\top)_{i_0}}{\mathrm{d} M_{i_1,j_1}}}_{n \times 1} \\
&= \exp(\underbrace{X}_{n \times d} \underbrace{(M \circ W)}_{d \times d} \underbrace{X^\top}_{d \times 1})_{i_0} \circ \underbrace{\frac{\mathrm{d}(X(M \circ W)X^\top)_{i_0}}{\mathrm{d} M_{i_1,j_1}}}_{n \times 1} \\
&= \underbrace{\widetilde{u}(M)_{i_0}}_{n \times 1} \circ \underbrace{\frac{\mathrm{d}(X(M \circ W)X^\top)_{i_0}}{\mathrm{d} M_{i_1,j_1}}}_{n \times 1} \\
&= \underbrace{\widetilde{u}(M)_{i_0}}_{n \times 1} \circ (\underbrace{W_{i_1,j_1}}_{\text{scalar}} \underbrace{X_{i_0,i_1}}_{\text{scalar}} \underbrace{X_{*,j_1}}_{n \times 1})
\end{aligned}
$$

where the first step follows from Definition C.1, the second step follows from Fact B.3, the third step follows from Definition C.1, and the fourth step follows from Lemma C.8. $\qquad\square$

### C.4   GRADIENT FOR EACH ENTRY OF $\widetilde{\alpha}(M)$

We introduce the Lemma of gradient for each entry of $\widetilde{\alpha}(M)$.

**Lemma C.10.** *If the following conditions hold:*

- *Let $\widetilde{u}(M)$ be defined in Definition C.1.*

- *Let $\widetilde{\alpha}(M)$ be defined in Definition C.2.*

*Let $i_1 \in [d]$, $j_1 \in [d]$, $i_0 \in [n]$, we have*

$$\underbrace{\frac{\mathrm{d}\widetilde{\alpha}(M)_{i_0}}{\mathrm{d}M_{i_1,j_1}}}_{\text{scalar}} = \langle \widetilde{u}(M)_{i_0} \circ (M_c)_{i_0}, W_{i_1,j_1} X_{i_0,i_1} X_{*,j_1} \rangle$$

*Proof.* We have

$$\underbrace{\frac{\mathrm{d}\widetilde{\alpha}(M)_{i_0}}{\mathrm{d}M_{i_1,j_1}}}_{\text{scalar}} = \underbrace{\frac{\mathrm{d}\langle (\widetilde{u}(M) \circ M_c)_{i_0}, \mathbf{1}_n \rangle}{\mathrm{d}M_{i_1,j_1}}}_{\text{scalar}}$$

$$= \langle \frac{\mathrm{d}(\widetilde{u}(M) \circ M_c)_{i_0}}{\mathrm{d}M_{i_1,j_1}}, \mathbf{1}_n \rangle$$

$$= \langle \frac{\mathrm{d}\widetilde{u}(M)_{i_0}}{\mathrm{d}M_{i_1,j_1}} \circ (M_c)_{i_0}, \mathbf{1}_n \rangle$$

$$= \langle \widetilde{u}(M)_{i_0} \circ (W_{i_1,j_1} X_{i_0,i_1} X_{*,j_1}) \circ (M_c)_{i_0}, \mathbf{1}_n \rangle$$

$$= \langle \widetilde{u}(M)_{i_0} \circ (M_c)_{i_0}, W_{i_1,j_1} X_{i_0,i_1} X_{*,j_1} \rangle$$

where the first step follows from Definition C.2, the second step follows from the product rule of inner product in Fact B.3, the third step follows from the product rule of Hadamard product in Fact B.3, the fourth step follows from Lemma C.9, and the last step follows from Fact B.4. □

## C.5 Gradient for Each Entry of $\widetilde{f}(M)$

We introduce the Lemma of the gradient for each entry of $\widetilde{f}(M)$.

**Lemma C.11.** *If the following conditions hold:*

- *Let $\widetilde{u}(M)$ be defined in Definition C.1.*

- *Let $\widetilde{\alpha}(M)$ be defined in Definition C.2.*

- *Let $\widetilde{f}(M)$ be defined in Definition C.3.*

*Let $i_1 \in [d]$, $j_1 \in [d]$, $i_0 \in [n]$, $j_0 \in [n]$, we have*

$$\frac{\mathrm{d}\widetilde{f}(M)_{i_0,j_0}}{\mathrm{d}M_{i_1,j_1}} = \widetilde{f}(M)_{i_0,j_0} W_{i_1,j_1} X_{i_0,i_1} X_{j_0,j_1} - \widetilde{f}(M)_{i_0,j_0} \langle \widetilde{f}(M)_{i_0}, W_{i_1,j_1} X_{i_0,i_1} X_{*,j_1} \rangle$$

*Proof.* We have

$$\frac{\mathrm{d}\widetilde{f}(M)_{i_0,j_0}}{\mathrm{d}M_{i_1,j_1}} = \frac{\mathrm{d}\widetilde{\alpha}(M)_{i_0}^{-1} (\widetilde{u}(M) \circ M_c)_{i_0,j_0}}{\mathrm{d}M_{i_1,j_1}}$$

$$= \frac{\mathrm{d}\widetilde{\alpha}(M)_{i_0}^{-1}}{\mathrm{d}M_{i_1,j_1}} (\widetilde{u}(M) \circ M_c)_{i_0,j_0} + \frac{\mathrm{d}(\widetilde{u}(M) \circ M_c)_{i_0,j_0}}{\mathrm{d}M_{i_1,j_1}} \widetilde{\alpha}(M)_{i_0}^{-1}$$

(4)

where the first step follows from Definition C.3, and the second step follows from Fact B.3.

In the following part, we compute the two terms separately.

For the first term above, we have

$$\frac{\mathrm{d}\widetilde{\alpha}(M)_{i_0}^{-1}}{\mathrm{d}M_{i_1,j_1}} (\widetilde{u}(M) \circ M_c)_{i_0,j_0}$$

$$
\begin{aligned}
&= (\widetilde{u}(M) \circ M_c)_{i_0,j_0}(-1)\widetilde{\alpha}(M)_{i_0}^{-2}\frac{\mathrm{d}\widetilde{\alpha}(M)_{i_0}}{\mathrm{d}M_{i_1,j_1}} \\
&= -(\widetilde{u}(M) \circ M_c)_{i_0,j_0}\langle \widetilde{u}(M)_{i_0} \circ (M_c)_{i_0}, W_{i_1,j_1}X_{i_0,i_1}X_{*,j_1}\rangle/\widetilde{\alpha}(M)_{i_0}^2 \\
&= -(\widetilde{\alpha}(M)_{i_0}^{-1}(M_c)_{i_0,j_0}\widetilde{u}(M)_{i_0,j_0})\langle \widetilde{\alpha}(M)_{i_0}^{-1}\widetilde{u}(M)_{i_0} \circ (M_c)_{i_0}, W_{i_1,j_1}X_{i_0,i_1}X_{*,j_1}\rangle \\
&= -\widetilde{f}(M)_{i_0,j_0}\langle \widetilde{f}(M)_{i_0}, W_{i_1,j_1}X_{i_0,i_1}X_{*,j_1}\rangle
\end{aligned}
\tag{5}
$$

where the first step follows from Fact B.3, the second step follows from Lemma C.10, the third step follows from basic algebra, and the fourth step follows from Definition C.3.

For the second term above, we have

$$
\begin{aligned}
&\frac{\mathrm{d}(\widetilde{u}(M) \circ M_c)_{i_0,j_0}}{\mathrm{d}M_{i_1,j_1}}\widetilde{\alpha}(M)_{i_0}^{-1} \\
&= \frac{\mathrm{d}\widetilde{u}(M)_{i_0,j_0}(M_c)_{i_0,j_0}}{\mathrm{d}M_{i_1,j_1}}\widetilde{\alpha}(M)_{i_0}^{-1} \\
&= (M_c)_{i_0,j_0}\left(\frac{\mathrm{d}\widetilde{u}(M)_{i_0}}{\mathrm{d}M_{i_1,j_1}}\right)_{j_0}\widetilde{\alpha}(M)_{i_0}^{-1} \\
&= ((M_c)_{i_0,j_0}\widetilde{u}(M)_{i_0,j_0}\widetilde{\alpha}(M)_{i_0}^{-1})W_{i_1,j_1}X_{i_0,i_1}X_{j_0,j_1} \\
&= \widetilde{f}(M)_{i_0,j_0}W_{i_1,j_1}X_{i_0,i_1}X_{j_0,j_1}
\end{aligned}
\tag{6}
$$

where the first step and the second step follow from basic algebra, the third step follows from Lemma C.9, and the fourth step follows from Definition C.3.

So, we have

$$
\begin{aligned}
\frac{\mathrm{d}\widetilde{f}(M)_{i_0,j_0}}{\mathrm{d}M_{i_1,j_1}} &= \frac{\mathrm{d}\widetilde{\alpha}(M)_{i_0}^{-1}}{\mathrm{d}M_{i_1,j_1}}(\widetilde{u}(M) \circ M_c)_{i_0,j_0} + \frac{\mathrm{d}(\widetilde{u}(M) \circ M_c)_{i_0,j_0}}{\mathrm{d}M_{i_1,j_1}}\widetilde{\alpha}(M)_{i_0}^{-1} \\
&= \widetilde{f}(M)_{i_0,j_0}W_{i_1,j_1}X_{i_0,i_1}X_{j_0,j_1} - \widetilde{f}(M)_{i_0,j_0}\langle \widetilde{f}(M)_{i_0}, W_{i_1,j_1}X_{i_0,i_1}X_{*,j_1}\rangle
\end{aligned}
$$

where the first step follows from Eq. (4), and the second step follows from Eq. (5) and Eq. (6). □

## C.6 GRADIENT FOR EACH ENTRY OF C(M)

We introduce the Lemma of gradient for each entry of $c(M)$.

**Lemma C.12.** *If the following conditions hold:*

- *Let $\widetilde{f}(M)$, $f$ be defined in Definition C.3.*

- *Let $c(M)$ be defined in Definition C.4.*

*Let $i_1 \in [d]$, $j_1 \in [d]$, $i_0 \in [n]$, $j_0 \in [n]$, we have*

$$
\frac{\mathrm{d}c(M)_{i_0,j_0}}{\mathrm{d}M_{i_1,j_1}} = \widetilde{f}(M)_{i_0,j_0}W_{i_1,j_1}X_{i_0,i_1}X_{j_0,j_1} - \widetilde{f}(M)_{i_0,j_0}\langle \widetilde{f}(M)_{i_0}, W_{i_1,j_1}X_{i_0,i_1}X_{*,j_1}\rangle
$$

*Proof.* We have

$$
\begin{aligned}
\frac{\mathrm{d}c(M)_{i_0,j_0}}{\mathrm{d}M_{i_1,j_1}} &= \frac{\mathrm{d}(\widetilde{f}(M)_{i_0,j_0} - f_{i_0,j_0})}{\mathrm{d}M_{i_1,j_1}} \\
&= \frac{\mathrm{d}\widetilde{f}(M)_{i_0,j_0}}{\mathrm{d}M_{i_1,j_1}} \\
&= \widetilde{f}(M)_{i_0,j_0}W_{i_1,j_1}X_{i_0,i_1}X_{j_0,j_1} - \widetilde{f}(M)_{i_0,j_0}\langle \widetilde{f}(M)_{i_0}, W_{i_1,j_1}X_{i_0,i_1}X_{*,j_1}\rangle
\end{aligned}
$$

where the first step follows from Definition C.4, the second step follows from Fact B.3, and the third step follows from Lemma C.11. □

## C.7 GRADIENT FOR $\mathcal{L}_{\mathrm{attn}}(M)$

We introduce the Lemma of the gradient for $\mathcal{L}_{\mathrm{attn}}(M)$.

**Lemma C.13.** *If the following conditions hold:*

- *Let $\widetilde{f}(M)$ be defined in Definition C.3.*

- *Let $c(M)$ be defined in Definition C.4.*

- *Let $\mathcal{L}_{\mathrm{attn}}(M)$ be defined in Definition C.5.*

*Let $i_1 \in [d]$, $j_1 \in [d]$, $i_0 \in [n]$, $j_0 \in [n]$, we have*

$$\frac{\mathrm{d}\mathcal{L}_{\mathrm{attn}}(M)}{\mathrm{d}M_{i_1,j_1}} = \sum_{i_0=1}^{n} \sum_{j_0=1}^{n} B_1(M) + B_2(M)$$

*where we have definitions:*

- $B_1(M) := c(M)_{i_0,j_0} \widetilde{f}(M)_{i_0,j_0} W_{i_1,j_1} X_{i_0,i_1} X_{j_0,j_1}$

- $B_2(M) := -c(M)_{i_0,j_0} \widetilde{f}(M)_{i_0,j_0} \langle \widetilde{f}(M)_{i_0}, W_{i_1,j_1} X_{i_0,i_1} X_{*,j_1} \rangle$

*Proof.* We have

$$\begin{aligned}
\frac{\mathrm{d}\mathcal{L}_{\mathrm{attn}}(M)}{\mathrm{d}M_{i_1,j_1}} &= \frac{1}{2} \frac{\mathrm{d}\|c(M)\|_F^2}{\mathrm{d}M_{i_1,j_1}} \\
&= \frac{1}{2} \frac{\mathrm{d}\sum_{i_0=1}^{n} \sum_{j_0=1}^{n} (c(M)_{i_0,j_0})^2}{\mathrm{d}M_{i_1,j_1}} \\
&= \frac{1}{2} \sum_{i_0=1}^{n} \sum_{j_0=1}^{n} \frac{\mathrm{d}(c(M)_{i_0,j_0})^2}{\mathrm{d}M_{i_1,j_1}} \\
&= \sum_{i_0=1}^{n} \sum_{j_0=1}^{n} c(M)_{i_0,j_0} \frac{\mathrm{d}c(M)_{i_0,j_0}}{\mathrm{d}M_{i_1,j_1}}
\end{aligned}$$

where the first step follows from Definition C.5, the second step follows from the definition of Frobenius's norm of the matrix, the third step follows from Fact B.3, and the fourth step follows from Fact B.3.

Following Lemma C.12, we have

$$\begin{aligned}
&\sum_{i_0=1}^{n} \sum_{j_0=1}^{n} c(M)_{i_0,j_0} \frac{\mathrm{d}c(M)_{i_0,j_0}}{\mathrm{d}M_{i_1,j_1}} \\
&= \sum_{i_0=1}^{n} \sum_{j_0=1}^{n} c(M)_{i_0,j_0} (\widetilde{f}(M)_{i_0,j_0} W_{i_1,j_1} X_{i_0,i_1} X_{j_0,j_1} - \widetilde{f}(M)_{i_0,j_0} \langle \widetilde{f}(M)_{i_0}, W_{i_1,j_1} X_{i_0,i_1} X_{*,j_1} \rangle) \\
&= c(M)_{i_0,j_0} \widetilde{f}(M)_{i_0,j_0} W_{i_1,j_1} X_{i_0,i_1} X_{j_0,j_1} - c(M)_{i_0,j_0} \widetilde{f}(M)_{i_0,j_0} \langle \widetilde{f}(M)_{i_0}, W_{i_1,j_1} X_{i_0,i_1} X_{*,j_1} \rangle \\
&:= \sum_{i_0=1}^{n} \sum_{j_0=1}^{n} B_1(M) + B_2(M)
\end{aligned}$$

where the second step follows from basic algebra. $\square$

## C.8 GRADIENT FOR $\mathcal{L}_{\mathrm{reg}}(M)$

We introduce the Lemma of the gradient for $\mathcal{L}_{\mathrm{reg}}(M)$.

**Lemma C.14.** *If the following conditions hold:*

- *Let $\mathcal{L}_{\mathrm{reg}}(M)$ be defined in Definition C.6.*

*Let $i_1 \in [d]$, $j_1 \in [d]$, we have*

$$\frac{\mathrm{d}\mathcal{L}_{\mathrm{reg}}(M)}{\mathrm{d}M_{i_1,j_1}} = B_3(M)$$

*where we have the definition:*

- $B_3(M) := \lambda M_{i_1,j_1}$

*Proof.* We have

$$\begin{aligned}
\frac{\mathrm{d}\mathcal{L}_{\mathrm{reg}}(M)}{\mathrm{d}M_{i_1,j_1}} &= \frac{1}{2}\lambda\frac{\mathrm{d}\|M\|_F^2}{\mathrm{d}M_{i_1,j_1}} \\
&= \frac{1}{2}\lambda(\frac{\mathrm{d}}{\mathrm{d}M_{i_1,j_1}}\sum_{i_0=1}^{d}\sum_{j_0=1}^{d}M_{i_0,j_0}^2) \\
&= \lambda M_{i_1,j_1} \\
&:= B_3(M)
\end{aligned}$$

where the first step follows from Definition C.6, the second step follows from the definition of Frobenius's norm of the matrix, and the third step follows from Fact B.3. $\qquad\square$

## C.9 GRADIENT FOR $\mathcal{L}(M)$

We introduce the Lemma of the gradient for $\mathcal{L}(M)$.

**Lemma C.15.** *If the following conditions hold:*

- *Let $\widetilde{u}(M)$ be defined in Definition C.1.*
- *Let $\widetilde{\alpha}(M)$ be defined in Definition C.2.*
- *Let $\widetilde{f}(M)$ be defined in Definition C.3.*
- *Let $\mathcal{L}_{\mathrm{attn}}(M)$ be defined in Definition C.5.*
- *Let $\mathcal{L}_{\mathrm{reg}}(M)$ be defined in Definition C.6.*
- *Let $\mathcal{L}(M)$ be defined in Definition C.7.*

*Let $i_1 \in [d]$, $j_1 \in [d]$, $i_0 \in [n]$, $j_0 \in [n]$, we have*

$$\frac{\mathrm{d}\mathcal{L}(M)}{\mathrm{d}M_{i_1,j_1}} = \sum_{i_0=1}^{n}\sum_{j_0=1}^{n}(B_1(M) + B_2(M)) + B_3(M)$$

*where we have definitions:*

- $B_1(M) := c(M)_{i_0,j_0}\widetilde{f}(M)_{i_0,j_0}W_{i_1,j_1}X_{i_0,i_1}X_{j_0,j_1}$
- $B_2(M) := -c(M)_{i_0,j_0}\widetilde{f}(M)_{i_0,j_0}\langle\widetilde{f}(M)_{i_0}, W_{i_1,j_1}X_{i_0,i_1}X_{*,j_1}\rangle$
- $B_3(M) := \lambda M_{i_1,j_1}$

*Proof.*

$$\begin{aligned}
\frac{\mathrm{d}\mathcal{L}(M)}{\mathrm{d}M_{i_1,j_1}} &= \frac{\mathrm{d}\mathcal{L}_{\mathrm{attn}}(M) + \mathcal{L}_{\mathrm{reg}}(M)}{\mathrm{d}M_{i_1,j_1}} \\
&= \frac{\mathrm{d}\mathcal{L}_{\mathrm{attn}}(M)}{\mathrm{d}M_{i_1,j_1}} + \frac{\mathrm{d}\mathcal{L}_{\mathrm{reg}}(M)}{\mathrm{d}M_{i_1,j_1}}
\end{aligned}$$

$$= \sum_{i_0=1}^{n} \sum_{j_0=1}^{n} (B_1(M) + B_2(M)) + B_3(M)$$

where the first step follows from Definition C.7, the second step follows from Fact B.3, and the third step follows from Lemma C.13 and Lemma C.14. $\qquad\square$

## D  MATRIX FORM

### D.1  MATRIX FORM OF $B(M)$

Given the matrix form, we define $p$ to simplify the calculation.

**Definition D.1.** *If the following conditions hold*

- *Let $X \in \mathbb{R}^{n \times d}$.*

- *Let $M \in [0,1]^{d \times d}$.*

- *Let $W \in \mathbb{R}^{d \times d}$.*

- *Let $c(M)$ be defined in Definition C.4.*

- *Let $\widetilde{f}(M)$ be defined in Definition C.3.*

*We define $p_1$ as follows*

$$p_1 := c(M) \circ \widetilde{f}(M)$$

*We define the $j_0$-th column of $p_1$ as follows*

$$(p_1)_{*,j_0} := (c(M) \circ \widetilde{f}(M))_{*,j_0}$$

*We define $p_2$ as follows*

$$p_2 := \mathrm{diag}(p_1 \cdot \mathbf{1}_n)\widetilde{f}(M)$$

*We define the $i_0$-th row of $p_2$ as follows*

$$(p_2)_{i_0} := \mathbf{1}_n^\top (p_1)_{i_0} \widetilde{f}(M)_{i_0} = \widetilde{f}(M)_{i_0} c(M)_{i_0}^\top \widetilde{f}(M)_{i_0}$$

*We define $p$ as follows*

$$p := p_1 - p_2 = c(M) \circ \widetilde{f}(M) - \mathrm{diag}((c(M) \circ \widetilde{f}(M)) \cdot \mathbf{1}_n)\widetilde{f}(M)$$

We introduce the matrix view of $B_1(M)$ and its summation.

**Lemma D.2** (Matrix view of $B_1(M)$)**.** *If we have the below conditions,*

- *Let $B_1(M, i_1, j_1) := c(M)_{i_0,j_0}\widetilde{f}(M)_{i_0,j_0} W_{i_1,j_1} X_{i_0,i_1} X_{j_0,j_1}$, which is defined in Lemma C.15*

- *We define $C_1(M) \in \mathbb{R}^{d \times d}$. For all $i_1, j_1 \in [d]$, let $C_1(i_1, j_1)$ denote the $(i_1, j_1)$-th entry of $C_1(M)$. We define $C_1(i_1, j_1) = B_1(M, i_1, j_1)$.*

*Then, we can show that*

- *Part 1. For $i_0, j_0 \in [n]$*

$$C_1(M) = \underbrace{c(M)_{i_0,j_0}\widetilde{f}(M)_{i_0,j_0}}_{\text{scalar}}(W \circ (X_{i_0} X_{j_0}^\top))$$

- *Part 2.*

$$\sum_{i_0=1}^{n} \sum_{j_0=1}^{n} C_1(M) = W \circ (X^\top p_1 X)$$

*Proof.* **Part 1.** We have

$$
\begin{aligned}
C_1(i_1, j_1) &= c(M)_{i_0,j_0} \widetilde{f}(M)_{i_0,j_0} W_{i_1,j_1} X_{i_0,i_1} X_{j_0,j_1} \\
&= \underbrace{c(M)_{i_0,j_0} \widetilde{f}(M)_{i_0,j_0}}_{\text{scalar}} \underbrace{(X_{i_0})_{i_1}}_{d \times 1} \underbrace{(W_{*,j_1})_{i_1}}_{d \times 1} \underbrace{(X_{j_0})_{j_1}}_{\text{scalar}} \\
&= \underbrace{c(M)_{i_0,j_0} \widetilde{f}(M)_{i_0,j_0}}_{\text{scalar}} \underbrace{(\operatorname{diag}(X_{i_0}) W_{*,j_1})_{i_1}}_{d \times 1} \underbrace{(X_{j_0})_{j_1}}_{\text{scalar}}
\end{aligned}
$$

where the first step follows from the definition of $C_1$, the second step follows from Fact B.1, and the third step follows from Fact B.2.

Following from Fact B.1, we can get $j_1$-th column of $C_1$

$$
\begin{aligned}
C_1(*, j_1) &= \underbrace{c(M)_{i_0,j_0} \widetilde{f}(M)_{i_0,j_0}}_{\text{scalar}} \underbrace{\operatorname{diag}(X_{i_0})}_{d \times d} \underbrace{W_{*,j_1}}_{d \times 1} \underbrace{(X_{j_0})_{j_1}}_{\text{scalar}} \\
&= \underbrace{c(M)_{i_0,j_0} \widetilde{f}(M)_{i_0,j_0}}_{\text{scalar}} \underbrace{\operatorname{diag}(X_{i_0})}_{d \times d} (\underbrace{W}_{d \times d} \underbrace{\operatorname{diag}(X_{j_0})}_{d \times d})_{*,j_1}
\end{aligned}
$$

where the second step follows from Fact B.2.

Following from Fact B.1, we can get $C_1(M)$

$$
\begin{aligned}
C_1(M) &= \underbrace{c(M)_{i_0,j_0} \widetilde{f}(M)_{i_0,j_0}}_{\text{scalar}} \underbrace{\operatorname{diag}(X_{i_0})}_{d \times d} \underbrace{W}_{d \times d} \underbrace{\operatorname{diag}(X_{j_0})}_{d \times d} \\
&= \underbrace{c(M)_{i_0,j_0} \widetilde{f}(M)_{i_0,j_0}}_{\text{scalar}} (W \circ (X_{i_0} X_{j_0}^\top))
\end{aligned} \tag{7}
$$

where the second step follows from Fact B.4.

**Part 2.** We further compute the summation of $C_1(M)$.

$$
\begin{aligned}
\sum_{i_0=1}^{n} \sum_{j_0=1}^{n} C_1(M) &= \sum_{i_0=1}^{n} \sum_{j_0=1}^{n} \underbrace{c(M)_{i_0,j_0} \widetilde{f}(M)_{i_0,j_0}}_{\text{scalar}} (W \circ X_{i_0} X_{j_0}^\top) \\
&= \sum_{i_0=1}^{n} \sum_{j_0=1}^{n} (W \circ (c(M)_{i_0,j_0} \widetilde{f}(M)_{i_0,j_0} X_{i_0} X_{j_0}^\top)) \\
&= W \circ \sum_{i_0=1}^{n} \sum_{j_0=1}^{n} c(M)_{i_0,j_0} \widetilde{f}(M)_{i_0,j_0} X_{i_0} X_{j_0}^\top \\
&= W \circ \sum_{j_0=1}^{n} \sum_{i_0=1}^{n} ((p_1)_{*,j_0})_{i_0} X_{i_0} X_{j_0}^\top
\end{aligned}
$$

where the first step follows from Eq. (7), the second step follows from basic algebra, the third step follows from Fact B.4, and the fourth step follows from Definition D.1.

Then following from Fact B.2, we have

$$
\begin{aligned}
W \circ \sum_{j_0=1}^{n} \sum_{i_0=1}^{n} ((p_1)_{*,j_0})_{i_0} X_{i_0} X_{j_0}^\top \\
= W \circ \sum_{j_0=1}^{n} X^\top (p_1)_{*,j_0} X_{j_0}^\top \\
= W \circ (X^\top p_1 X)
\end{aligned}
$$

$\square$

We introduce the matrix view of $B_2(M)$ and its summation.

**Lemma D.3** (Matrix view of $B_2(M)$). *If we have the below conditions,*

- *Let* $B_2(M, i_1, j_1) := -c(M)_{i_0,j_0} \widetilde{f}(M)_{i_0,j_0} \langle \widetilde{f}(M)_{i_0}, W_{i_1,j_1} X_{i_0,i_1} X_{*,j_1} \rangle$ *be defined in Lemma C.15.*

- *We define* $C_2(M) \in \mathbb{R}^{d \times d}$. *For all* $i_1, j_1 \in [d]$, *let* $C_2(i_1, j_1)$ *denote the* $(i_1, j_1)$-*th entry of* $C_2(M)$. *We define* $C_2(i_1, j_1) = B_2(M, i_1, j_1)$.

*Then, we can show that*

- *Part 1. For* $i_0, j_0 \in [n]$

$$C_2(M) = -c(M)_{i_0,j_0} \widetilde{f}(M)_{i_0,j_0} \underbrace{\mathrm{diag}(X_{i_0})}_{d \times d} \underbrace{W}_{d \times d} \underbrace{\mathrm{diag}(X^\top \widetilde{f}(M)_{i_0})}_{d \times d}$$

- *Part 2.*

$$\sum_{i_0=1}^{n} \sum_{j_0=1}^{n} C_2(M) = -W \circ (X^\top p_2 X)$$

*Proof.* **Part 1.** We have

$$-C_2(i_1, j_1) = c(M)_{i_0,j_0} \widetilde{f}(M)_{i_0,j_0} \langle \widetilde{f}(M)_{i_0}, W_{i_1,j_1} X_{i_0,i_1} X_{*,j_1} \rangle$$

$$= c(M)_{i_0,j_0} \widetilde{f}(M)_{i_0,j_0} \underbrace{\widetilde{f}(M)_{i_0}^\top}_{1 \times n} \underbrace{X_{*,j_1} W_{i_1,j_1} X_{i_0,i_1}}_{n \times 1}$$

$$= c(M)_{i_0,j_0} \widetilde{f}(M)_{i_0,j_0} \underbrace{\widetilde{f}(M)_{i_0}^\top}_{1 \times n} \underbrace{X_{*,j_1} (X_{i_0})_{i_1} (W_{*,j_1})_{i_1}}_{n \times 1}$$

$$= c(M)_{i_0,j_0} \widetilde{f}(M)_{i_0,j_0} \underbrace{\widetilde{f}(M)_{i_0}^\top}_{1 \times n} \underbrace{X_{*,j_1} (\mathrm{diag}(X_{i_0}) W_{*,j_1})_{i_1}}_{n \times 1}$$

where the first step follows from the definition of $C_2$, the second step, the third step, and the fourth step follow from Fact B.4.

Following from Fact B.1, we can get $j_1$-th column of $C_2$

$$-C_2(*, j_1) = \underbrace{\mathrm{diag}(X_{i_0})}_{d \times d} \underbrace{W_{*,j_1}}_{d \times 1} \underbrace{c(M)_{i_0,j_0} \widetilde{f}(M)_{i_0,j_0} \widetilde{f}(M)_{i_0}^\top}_{1 \times n} \underbrace{X_{*,j_1}}_{n \times 1}$$

$$= c(M)_{i_0,j_0} \widetilde{f}(M)_{i_0,j_0} \underbrace{\mathrm{diag}(X_{i_0})}_{d \times d} \underbrace{W_{*,j_1}}_{d \times 1} \underbrace{\widetilde{f}(M)_{i_0}^\top}_{1 \times n} \underbrace{X_{*,j_1}}_{n \times 1}$$

$$= c(M)_{i_0,j_0} \widetilde{f}(M)_{i_0,j_0} \underbrace{\mathrm{diag}(X_{i_0})}_{d \times d} \underbrace{W_{*,j_1}}_{d \times 1} \underbrace{(X^\top \widetilde{f}(M)_{i_0})_{j_1}}_{\text{scalar}}$$

$$= c(M)_{i_0,j_0} \widetilde{f}(M)_{i_0,j_0} \underbrace{\mathrm{diag}(X_{i_0})}_{d \times d} \underbrace{(W \, \mathrm{diag}(X^\top \widetilde{f}(M)_{i_0}))}_{d \times d} {}_{*,j_1}$$

where the second step and the fourth step follows from Fact B.4, and the third step follows from Fact B.1.

Following from Fact B.1, we can get $C_2$.

$$-C_2(M) = c(M)_{i_0,j_0} \widetilde{f}(M)_{i_0,j_0} \underbrace{\mathrm{diag}(X_{i_0})}_{d \times d} \underbrace{W}_{d \times d} \underbrace{\mathrm{diag}(X^\top \widetilde{f}(M)_{i_0})}_{d \times d} \tag{8}$$

**Part 2.** We further compute the summation of $C_2$

$$-\sum_{i_0=1}^{n} \sum_{j_0=1}^{n} C_2(M) = \sum_{i_0=1}^{n} \sum_{j_0=1}^{n} c(M)_{i_0,j_0} \widetilde{f}(M)_{i_0,j_0} \underbrace{\mathrm{diag}(X_{i_0})}_{d \times d} \underbrace{W}_{d \times d} \underbrace{\mathrm{diag}(X^\top \widetilde{f}(M)_{i_0})}_{d \times d}$$

$$= \sum_{i_0=1}^{n} \sum_{j_0=1}^{n} c(M)_{i_0,j_0} \widetilde{f}(M)_{i_0,j_0} ((X_{i_0} \widetilde{f}(M)_{i_0}^{\top} X) \circ W)$$

$$= W \circ \sum_{i_0=1}^{n} (X_{i_0} \widetilde{f}(M)_{i_0}^{\top} X) \sum_{j_0=1}^{n} ((p_1)_{i_0})_{j_0}$$

where the first step follows from Eq. (8), the second step and the third step follow from Fact B.4.
Following from Fact B.2, we have

$$W \circ \sum_{i_0=1}^{n} (X_{i_0} \widetilde{f}(M)_{i_0}^{\top} X) \sum_{j_0=1}^{n} ((p_1)_{i_0})_{j_0}$$

$$= W \circ \sum_{i_0=1}^{n} (X_{i_0} \widetilde{f}(M)_{i_0}^{\top} X) \mathbf{1}_n^{\top} (p_1)_{i_0}$$

$$= W \circ (X^{\top} \operatorname{diag}(p_1 \cdot \mathbf{1}_n) \widetilde{f}(M) X)$$

$$= W \circ (X^{\top} p_2 X)$$

where the third step follows from Definition D.1. $\qquad \square$

We introduce the matrix view of $B_3(M)$.

**Lemma D.4** (Matrix view of $B_3(M)$). *If the following conditions hold*

- *Let $B_3(M, i_1, j_1) := \lambda M_{i_1, j_1}$ be defined in Lemma C.15.*

- *We define $C_3(M) \in \mathbb{R}^{d \times d}$. For all $i_1, j_1 \in [d]$, let $C_3(i_1, j_1)$ denote the $(i_1, j_1)$-th entry of $C_3(M)$. We define $C_3(i_1, j_1) = B_3(M, i_1, j_1)$.*

*We can show that*

$$C_3(M) = \lambda M.$$

*Proof.* The proof is straightforward. By the definition of $C_3(M)$, for all $i_1, j_1 \in [d]$, the $(i_1, j_1)$-th entry of $C_3(M)$ is given by $C_3(i_1, j_1) = B_3(M, i_1, j_1) = \lambda M_{i_1, j_1}$. Thus, the entire matrix $C_3(M)$ has entries that correspond to those of $\lambda M$. Therefore, we can conclude that $C_3(M) = \lambda M$ as required. $\qquad \square$

### D.2 MATRIX FORM OF $\frac{\mathrm{d}}{\mathrm{d}M} \mathcal{L}(M)$

We introduce the matrix form of the overall loss function.

**Theorem D.5** (Close form of gradient, formal version of Theorem 5.3). *If the following conditions hold*

- *Let $\mathcal{L}(M)$ be defined in Definition C.7.*

- *Let $p$ be defined in Definition D.1.*

- *Let $X \in \mathbb{R}^{n \times d}$.*

- *Let $M \in [0, 1]^{d \times d}$.*

- *Let $W \in \mathbb{R}^{d \times d}$.*

*We can show that*

$$\frac{\mathrm{d}\mathcal{L}(M)}{\mathrm{d}M} = W \circ (X^{\top} p X) + \lambda M.$$

*Proof.* We have

$$
\begin{aligned}
\frac{\mathrm{d}\mathcal{L}(M)}{\mathrm{d}M} &= \sum_{i_0=1}^{n} \sum_{j_0=1}^{n} (C_1(M) + C_2(M)) + C_3(M) \\
&= W \circ (X^\top p_1 X) - W \circ (X^\top p_2 X) + \lambda M \\
&= W \circ (X^\top (p_1 - p_2) X) + \lambda M \\
&= W \circ (X^\top p X) + \lambda M
\end{aligned}
$$

where the first step follows from Lemma C.15, the second step follows from Lemma D.2, Lemma D.3, and Lemma D.4, the third step follows from basic algebra, and the fourth step follows from Definition D.1. □

# E    BOUNDS FOR BASIC FUNCTIONS

## E.1    BASIC ASSUMPTIONS

Here, we introduce our bounded parameters assumption.

**Assumption E.1** (Bounded parameters). *We assume the following conditions*

- *Let $R$ be some fixed constant satisfies $R > 1$.*

- *Let $X \in \mathbb{R}^{n \times d}, W \in \mathbb{R}^{d \times d}$. We have $\|X\|_F \leq R$ and $\|W\|_F \leq R$.*

Here, we present the lemma of bounds for $M$ and $M_c$.

**Lemma E.2** (Bounds for $M$ and $M_c$). *Let $M \in [0,1]^{d \times d}$ and $M_c \in \{0,1\}^{n \times n}$ be the causal attention mask defined in Definition 3.1. For $M$, we have*

$$
\|M\|_F \leq d
$$

*For $M_c$, we have*

$$
\|M_c\|_F \leq n
$$

*Proof.* This Lemma simply follows from the definition of the Frobenius norm, given that the max value of each entry in $M$ and $M_c$ is 1. □

## E.2    BOUNDS FOR BASIC FUNCTIONS

We first introduce the lemma of bounds for basic functions.

**Lemma E.3.** *Under Assumption E.1, for all $i_0 \in [n]$, $j_0 \in [n]$, $i_1 \in [d]$, $j_1 \in [d]$, we have the following bounds*

- *Part 1.*

$$
\|\widetilde{f}(M)\|_F \leq \sqrt{n}
$$

- *Part 2.*

$$
\|c(M)\|_F \leq 2\sqrt{n}
$$

- *Part 3.*

$$
\|(c(M) \circ \widetilde{f}(M))\|_F \leq 2\sqrt{n}
$$

- *Part 4.*

$$
|\widetilde{f}(M)_{i_0,j_0}| \leq 1
$$

- *Part 5.*

$$|W_{i_1,j_1}| \le R$$

- *Part 6.*

$$|X_{i_0,i_1}| \le R$$

- *Part 7.*

$$\|\widetilde{f}(M)_{i_0}\|_2 \le 1$$

- *Part 8.*

$$|\widetilde{f}(M)_{i_0,j_0} W_{i_1,j_1} X_{i_0,i_1} X_{j_0,j_1}| \le R^3$$

- *Part 9.*

$$|\widetilde{f}(M)_{i_0,j_0} \langle \widetilde{f}(M)_{i_0}, W_{i_1,j_1} X_{i_0,i_1} X_{*,j_1} \rangle| \le R^3$$

- *Part 10.*

$$\| \operatorname{diag}((c(M) \circ \widetilde{f}(M)) \cdot \mathbf{1}_n) \|_F \le 2n$$

*Proof.* **Proof of Part 1.** Each entry in $\widetilde{f}(M)$ present a probability, thus for $i_0 \in [n]$, $j_0 \in [n]$, we have

$$0 \le \widetilde{f}(M)_{i_0,j_0} \le 1.$$

For any $i_0$-th row of $\widetilde{f}(M)$, following from the definition of Softmax function, we know

$$\sum_{j_0=1}^{n} \widetilde{f}(M)_{i_0,j_0} = 1.$$

So we have

$$\sum_{j_0=1}^{n} \widetilde{f}(M)_{i_0,j_0}^2 \le 1$$

which follows from $\widetilde{f}(M)_{i_0,j_0} \le (\widetilde{f}(M)_{i_0,j_0})^2$. Then, we can show

$$\|\widetilde{f}(M)\|_F = \sqrt{\sum_{i_0=1}^{n} \sum_{j_0=1}^{n} \widetilde{f}(M)_{i_0,j_0}^2} \le \sqrt{n}$$

**Proof of Part 2.** Following from **Part 1.**, we can show

$$\|\widetilde{f}(M)\|_F \le \sqrt{n}$$

and

$$\|f\|_F \le \sqrt{n}.$$

Then we have

$$\begin{aligned}
\|c(M)\|_F &= \|\widetilde{f}(M) - f\|_F \\
&\le \|\widetilde{f}(M)\|_F + \|f\|_F \\
&\le 2\sqrt{n}
\end{aligned}$$

where the first step follows from Definition C.4, the second step follows the triangle inequality.

**Proof of Part 3.** We have $0 \le \widetilde{f}(M)_{i_0,j_0} \le 1$, so we have

$$\|(c(M) \circ \widetilde{f}(M))\|_F \le \|c(M)\|_F$$

$$\leq 2\sqrt{n}$$

where the second step follows from **Part 2.**.

**Proof of Part 4.** See **Proof of Part 1.**.

**Proof of Part 5.** The proof simply follows from Assumption E.1 and Fact B.5.

**Proof of Part 6.** The proof simply follows from Assumption E.1 and Fact B.5.

**Proof of Part 7.** See **Proof of Part 1.**.

**Proof of Part 8.** The proof simply follows from **Part 4.**, **Part 5.**, **Part 6.** and **Part 7.**.

**Proof of Part 9.** The proof simply follows from **Part 4.**, **Part 5.**, **Part 6.** and **Part 7.**.

**Proof of Part 10.** We have

$$
\begin{aligned}
\| \operatorname{diag}((c(M) \circ \widetilde{f}(M)) \cdot \mathbf{1}_n) \|_F &= \|(c(M) \circ \widetilde{f}(M)) \cdot \mathbf{1}_n\|_2 \\
&\leq \|\mathbf{1}_n\|_2 \|(c(M) \circ \widetilde{f}(M))\|_F \\
&= \sqrt{n} \cdot 2\sqrt{n} \\
&= 2n
\end{aligned}
$$

where the first step follows from Fact B.4 the second step follows from Fact B.5, the third step follows from **Part 3.**, and the last step follows from simple algebra. □

### E.3 Bounds for Gradient of $\widetilde{f}(M)$

We introduce the lemma of bounds for the gradient of $\widetilde{f}(M)$.

**Lemma E.4.** *If the following conditions hold*

- *Let $\widetilde{f}(M)$ be defined in Definition C.3.*

- *Assumption E.1 holds.*

*Then we have*

$$\|\frac{\mathrm{d}\operatorname{vec}(\widetilde{f}(M))}{\mathrm{d}\operatorname{vec}(M)}\|_F \leq 2dnR^3$$

*Proof.* We have

$$
\begin{aligned}
&|\frac{\mathrm{d}f(M)_{i_0,j_0}}{\mathrm{d}M_{i_1,j_1}}| \\
&= |\widetilde{f}(M)_{i_0,j_0} W_{i_1,j_1} X_{i_0,i_1} X_{j_0,j_1} - \widetilde{f}(M)_{i_0,j_0} \langle \widetilde{f}(M)_{i_0}, W_{i_1,j_1} X_{i_0,i_1} X_{*,j_1}\rangle| \\
&\leq |\widetilde{f}(M)_{i_0,j_0} W_{i_1,j_1} X_{i_0,i_1} X_{j_0,j_1}| + |\widetilde{f}(M)_{i_0,j_0} \langle \widetilde{f}(M)_{i_0}, W_{i_1,j_1} X_{i_0,i_1} X_{*,j_1}\rangle| \\
&\leq 2R^3
\end{aligned}
$$

For $\frac{\mathrm{d}\operatorname{vec}(\widetilde{f}(M))}{\mathrm{d}\operatorname{vec}(M)}$, we can show

$$
\begin{aligned}
\|\frac{\mathrm{d}\operatorname{vec}(\widetilde{f}(M))}{\mathrm{d}\operatorname{vec}(M)}\|_F &= \sqrt{\sum_{i_2=1}^{n^2}\sum_{j_2=1}^{d^2} |\frac{\mathrm{d}\operatorname{vec}(\widetilde{f}(M))_{i_0}}{\mathrm{d}\operatorname{vec}(M)_{j_0}}|} \\
&\leq 2ndR^3
\end{aligned}
$$

□

# F  LIPSCHITZ OF GRADIENT

## F.1  USEFUL FACTS

Here, we introduce the fact of the mean value theorem for matrix function.

**Fact F.1** (Mean value theorem for matrix function, Fact C.6 in Liang et al. (2024e)). *If the following conditions hold*

- *Let $X, Y \in C \subset \mathbb{R}^{d \times d}$ where $C$ is an open convex domain.*
- *Let $g(X) : C \to \mathbb{R}^{n \times n}$ be a differentiable matrix function on $C$.*
- *Let $\|\frac{d \operatorname{vec}(g(X))}{d \operatorname{vec}(X)}\|_F \leq R$ for all $x \in C$.*

*We have*

$$\|g(Y) - g(X)\|_F \leq R\|Y - X\|_F.$$

*Proof.* For the convenience of proof, we define $x$ and $y$ as follows:

- $x := \operatorname{vec}(X)$ and $y := \operatorname{vec}(Y)$.
- $h(x) := \operatorname{vec}(g(X))$ and $h(y) := \operatorname{vec}(g(Y))$.
- $h'(a)$ denotes a matrix which its $(i, j)$-th term is $\frac{dh(a)_j}{da_i}$.

Assume we have 1-variable function $\gamma(c) = f(x + c(y - x))$, we can apply Mean Value Theorem:

$$f(y) - f(x) = \gamma(1) - \gamma(0) = \gamma'(t)(1 - 0) = \nabla f(x + t(y - x))^\top (y - x) \tag{9}$$

where $t \in [0, 1]$. Let $G(c) := (h(y) - h(x))^\top h(c)$, we have

$$
\begin{aligned}
\|g(Y) - g(X)\|_F^2 &= G(y) - G(x) \\
&= \nabla G(x + t(y - x))^\top (y - x) \\
&= (\underbrace{h'(x + t(y - x))}_{d^2 \times n^2} \cdot \underbrace{h(y) - h(x)}_{n^2 \times 1})^\top \cdot (y - x) \\
&\leq \|h'(x + t(y - x))\| \cdot \|h(y) - h(x)\|_2 \cdot \|y - x\|_2
\end{aligned}
$$

where the second step follows from Eq. (9), the third step follows from the chain rule, the fourth step follows from the Cauchy-Schwartz inequality.

By definition of matrix Frobenius norm and vector $\ell_2$ norm, we have

$$\|g(Y) - g(X)\|_F = \|h(y) - h(x)\|_2$$

and

$$\|Y - X\|_F = \|y - x\|_2$$

so we can show

$$\|g(Y) - g(X)\|_F \leq R\|Y - X\|_F$$

which follows from $\|\frac{d \operatorname{vec}(g(X))}{d \operatorname{vec}(X)}\|_F \leq R$ for all $x \in C$. $\square$

Here, we introduce the fact of Lipschitz for the product of functions.

**Fact F.2** (Lipschitz for product of functions, Fact H.2 in Deng et al. (2023)). *Under following conditions*

- *Let $\{f_i(x)\}_{i=1}^n$ be a sequence of functions with the same domain and range.*

- *For each $i \in [n]$, we have*

  - *$f_i(x)$ is bounded: $\forall x, \|f_i(x)\|_F \leq R_i$ with $R_i \geq 1$.*
  - *$f_i(x)$ is Lipschitz continuous: $\forall x, y, \|f_i(x) - f_i(y)\|_F \leq L_i \|x - y\|_F$.*

*Then we have*

$$\|\prod_{i=1}^{n} f_i(x) - \prod_{i=1}^{n} f_i(y)\|_F \leq 2^{n-1} \cdot \max_{i \in [n]}\{L_i\} \cdot (\prod_{i=1}^{n} R_i) \cdot \|x - y\|_F$$

## F.2 LIPSCHITZ OF $\widetilde{f}(M)$

We introduce the lemma about Lipschitz of $\widetilde{f}(M)$.

**Lemma F.3** (Lipschitz of $\widetilde{f}(M)$). *Under the following conditions*

- *Assumption E.1 holds.*

- *Let $\widetilde{f}(M)$ be defined as Definition C.3.*

*For $M, \widetilde{M} \in \mathbb{R}^{d \times d}$, we have*

$$\|\widetilde{f}(M) - \widetilde{f}(\widetilde{M})\|_F \leq 2dnR^3 \|M - \widetilde{M}\|_F$$

*Proof.* We have

$$\|\widetilde{f}(M) - \widetilde{f}(\widetilde{M})\|_F \leq \|\nabla \widetilde{f}(M)\|_F \cdot \|M - \widetilde{M}\|_F$$
$$\leq 2dnR^3 \cdot \|M - \widetilde{M}\|_F$$

where the first step follows from Fact F.1, the second step follows from Lemma E.4. $\square$

## F.3 LIPSCHITZ OF $c(M)$

We introduce the lemma about Lipschitz of $c(M)$.

**Lemma F.4** (Lipschitz of $c(M)$). *Under the following conditions*

- *Assumption E.1 holds.*

- *Let $c(M)$ be defined as Definition C.4.*

*For $M, \widetilde{M} \in \mathbb{R}^{d \times d}$, we have*

$$\|c(M) - c(\widetilde{M})\|_F \leq 2dnR^3 \|M - \widetilde{M}\|_F$$

*Proof.* We have

$$\|c(M) - c(\widetilde{M})\|_F \leq \|\nabla c(M)\|_F \cdot \|M - \widetilde{M}\|_F$$
$$= \|\nabla \widetilde{f}(M)\|_F \cdot \|M - \widetilde{M}\|_F$$
$$\leq 2dnR^3 \cdot \|M - \widetilde{M}\|_F$$

where the first step follows from Fact F.1, the second step follows from Lemma C.12, the third step follows from Lemma E.4. $\square$

## F.4 LIPSCHITZ OF $\widetilde{f}(M) \circ c(M)$

We introduce the lemma about Lipschitz of $\widetilde{f}(M) \circ c(M)$.

**Lemma F.5** (Lipschitz of $\widetilde{f}(M) \circ c(M)$). *Under the following conditions*

- *Assumption E.1 holds.*

- *Let $c(M)$ be defined as Definition C.4.*

- *Let $\widetilde{f}(M)$ be defined as Definition C.3.*

*For $M, \widetilde{M} \in \mathbb{R}^{d \times d}$, we have*

$$\|\widetilde{f}(M) \circ c(M) - \widetilde{f}(\widetilde{M}) \circ c(\widetilde{M})\|_F \leq 6dn^{3/2}R^3\|M - \widetilde{M}\|_F$$

*Proof.* We have

$$\begin{aligned}
\text{LHS} &\leq \|\widetilde{f}(M) \circ c(M) - \widetilde{f}(M) \circ c(\widetilde{M})\|_F + \|\widetilde{f}(M) \circ c(\widetilde{M}) - \widetilde{f}(\widetilde{M}) \circ c(\widetilde{M})\|_F \\
&\leq \|\widetilde{f}(M)\|_F \cdot \|c(M) - c(\widetilde{M})\|_F + \|c(\widetilde{M})\|_F \cdot \|\widetilde{f}(M) - \widetilde{f}(\widetilde{M})\|_F \\
&\leq \sqrt{n} \cdot \|c(M) - c(\widetilde{M})\|_F + 2\sqrt{n} \cdot \|\widetilde{f}(M) - \widetilde{f}(\widetilde{M})\|_F \\
&\leq \sqrt{n} \cdot 2dnR^3\|M - \widetilde{M}\|_F + 2\sqrt{n} \cdot 2dnR^3\|M - \widetilde{M}\|_F
\end{aligned}$$

where the first step follows from triangle inequality, the second step follows from Fact B.5, the third step follows from Lemma E.3, the fourth step follows from Lemma F.4 and Lemma F.3.

So we have

$$\|\widetilde{f}(M) \circ c(M) - \widetilde{f}(\widetilde{M}) \circ c(\widetilde{M})\|_F \leq 6dn^{3/2}R^3\|M - \widetilde{M}\|_F$$

$\square$

## F.5 Lipschitz of $\mathrm{diag}((\widetilde{f}(M) \circ c(M)) \cdot \mathbf{1}_n)$

We introduce the lemma about Lipschitz of $\mathrm{diag}((\widetilde{f}(M) \circ c(M)) \cdot \mathbf{1}_n)$.

**Lemma F.6** (Lipschitz of $\mathrm{diag}((\widetilde{f}(M) \circ c(M)) \cdot \mathbf{1}_n)$)**.** *If the following conditions hold*

- *Assumption E.1 holds.*

- *Let $c(M)$ be defined as Definition C.4.*

- *Let $\widetilde{f}(M)$ be defined as Definition C.3.*

*For $M, \widetilde{M} \in \mathbb{R}^{d \times d}$, we have*

$$\|\mathrm{diag}((\widetilde{f}(M) \circ c(M)) \cdot \mathbf{1}_n) - \mathrm{diag}((\widetilde{f}(\widetilde{M}) \circ c(\widetilde{M})) \cdot \mathbf{1}_n)\|_F \leq 6dn^2R^3\|M - \widetilde{M}\|_F$$

*Proof.* We have

$$\begin{aligned}
\text{LHS} &= \|(\widetilde{f}(M) \circ c(M)) \cdot \mathbf{1}_n - (\widetilde{f}(\widetilde{M}) \circ c(\widetilde{M})) \cdot \mathbf{1}_n\|_2 \\
&= \|((\widetilde{f}(M) \circ c(M)) - (\widetilde{f}(\widetilde{M}) \circ c(\widetilde{M}))) \cdot \mathbf{1}_n\|_2 \\
&\leq \|\widetilde{f}(M) \circ c(M) - \widetilde{f}(\widetilde{M}) \circ c(\widetilde{M})\| \cdot \|\mathbf{1}_n\|_2 \\
&= \sqrt{n}\|\widetilde{f}(M) \circ c(M) - \widetilde{f}(\widetilde{M}) \circ c(\widetilde{M})\| \qquad (10)
\end{aligned}$$

where the first step follows from Fact B.4, the second step follows from basic algebra, the third step follows from Fact B.5, and the fourth step follows from $\|\mathbf{1}_n\|_2 = \sqrt{n}$.

Then we have

$$\|\widetilde{f}(M) \circ c(M) - \widetilde{f}(\widetilde{M}) \circ c(\widetilde{M})\| \leq \|\widetilde{f}(M) \circ c(M) - \widetilde{f}(\widetilde{M}) \circ c(\widetilde{M})\|_F \qquad (11)$$

which follows from Fact B.5.

Following Eq. (10), Eq. (11) and Lemma F.5, we have

$$\text{LHS} \leq \sqrt{n} \cdot 6dn^{3/2}R^3\|M - \widetilde{M}\|_F = 6dn^2R^3\|M - \widetilde{M}\|_F$$

$\square$

## F.6 Lipschitz of $\mathrm{diag}((\widetilde{f}(M) \circ c(M)) \cdot \mathbf{1}_n)\widetilde{f}(M)$

We introduce the lemma about Lipschitz of $\mathrm{diag}((\widetilde{f}(M) \circ c(M)) \cdot \mathbf{1}_n)\widetilde{f}(M)$.

**Lemma F.7** (Lipschitz of $\mathrm{diag}((\widetilde{f}(M) \circ c(M)) \cdot \mathbf{1}_n)\widetilde{f}(M)$). *If the following conditions hold*

- *Assumption E.1 holds.*

- *Let $c(M)$ be defined as Definition C.4.*

- *Let $\widetilde{f}(M)$ be defined as Definition C.3.*

*For $M, \widetilde{M} \in \mathbb{R}^{d \times d}$, we have*

$$\| \mathrm{diag}((\widetilde{f}(M) \circ c(M)) \cdot \mathbf{1}_n)\widetilde{f}(M) - \mathrm{diag}((\widetilde{f}(\widetilde{M}) \circ c(\widetilde{M})) \cdot \mathbf{1}_n)\widetilde{f}(\widetilde{M})\|_F \leq 24dn^{7/2}R^3\|M - \widetilde{M}\|_F$$

*Proof.* Following Fact F.2, we have

$$\| \mathrm{diag}((\widetilde{f}(M) \circ c(M)) \cdot \mathbf{1}_n)\widetilde{f}(M) - \mathrm{diag}((\widetilde{f}(\widetilde{M}) \circ c(\widetilde{M})) \cdot \mathbf{1}_n)\widetilde{f}(\widetilde{M})\|_F$$
$$\leq 2^1 \cdot \max\{6dn^2R^3, 6dn^{3/2}R^3\} \cdot (\sqrt{n} \cdot 2n)\|M - \widetilde{M}\|_F$$
$$= 24dn^{7/2}R^3\|M - \widetilde{M}\|_F$$

where we have the upper bound in Lemma E.3, the Lipschitz of $\mathrm{diag}((\widetilde{f}(M) \circ c(M)))$ and $\widetilde{f}(M)$ in Lemma F.3 and Lemma F.6. $\square$

## F.7 Lipschitz of Gradient

We introduce the lemma about Lipschitz of the gradient.

**Theorem F.8** (Lipschitz of the gradient, formal version of Theorem 5.4). *We can show $\nabla_M \mathcal{L}(M)$ is L-Lipschitz.*

*If the following conditions hold*

- *Assumption E.1 holds.*

- *Let $c(M)$ be defined as Definition C.4.*

- *Let $\widetilde{f}(M)$ be defined as Definition C.3.*

*For $M, \widetilde{M} \in \mathbb{R}^{d \times d}$, we have*

$$\|\nabla_M \mathcal{L}(M) - \nabla_M \mathcal{L}(\widetilde{M})\|_F \leq (\lambda + 30dn^{7/2}R^6) \cdot \|M - \widetilde{M}\|_F$$

*Proof.* We have

$$\|\nabla_M \mathcal{L}(M) - \nabla_M \mathcal{L}(\widetilde{M})\|_F$$
$$= \|W \circ (X^\top(c(M) \circ \widetilde{f}(M) - \mathrm{diag}(c(M) \circ \widetilde{f}(M) \cdot \mathbf{1}_n)\widetilde{f}(M)$$
$$- c(\widetilde{M}) \circ \widetilde{f}(\widetilde{M}) + \mathrm{diag}(c(\widetilde{M}) \circ \widetilde{f}(\widetilde{M}) \cdot \mathbf{1}_n)\widetilde{f}(\widetilde{M}))X) + \lambda M - \lambda \widetilde{M}\|_F$$
$$\leq \|W \circ (X^\top(c(M) \circ \widetilde{f}(M) - \mathrm{diag}(c(M) \circ \widetilde{f}(M) \cdot \mathbf{1}_n)\widetilde{f}(M)$$
$$- c(\widetilde{M}) \circ \widetilde{f}(\widetilde{M}) + \mathrm{diag}(c(\widetilde{M}) \circ \widetilde{f}(\widetilde{M}) \cdot \mathbf{1}_n)\widetilde{f}(\widetilde{M}))X)\|_F + \|\lambda(M - \widetilde{M})\|_F \qquad (12)$$

where the first step follows from Theorem D.5, and the second step follows the triangle inequality. Now, we proof these two terms separately.

For the first term, we have

$$\|W \circ (X^\top(c(M) \circ \widetilde{f}(M) - \mathrm{diag}(c(M) \circ \widetilde{f}(M) \cdot \mathbf{1}_n)\widetilde{f}(M)$$
$$- c(\widetilde{M}) \circ \widetilde{f}(\widetilde{M}) + \mathrm{diag}(c(\widetilde{M}) \circ \widetilde{f}(\widetilde{M}) \cdot \mathbf{1}_n)\widetilde{f}(\widetilde{M}))X)\|_F$$

$$
\begin{aligned}
&\leq \|W\|_F \cdot \|X^\top(c(M) \circ \widetilde{f}(M) - \mathrm{diag}(c(M) \circ \widetilde{f}(M) \cdot \mathbf{1}_n)\widetilde{f}(M) \\
&\quad - c(\widetilde{M}) \circ \widetilde{f}(\widetilde{M}) + \mathrm{diag}(c(\widetilde{M}) \circ \widetilde{f}(\widetilde{M}) \cdot \mathbf{1}_n)\widetilde{f}(\widetilde{M}))X\|_F \\
&\leq \|W\|_F \cdot \|X\|_F^2 \cdot \|c(M) \circ \widetilde{f}(M) - \mathrm{diag}(c(M) \circ \widetilde{f}(M) \cdot \mathbf{1}_n)\widetilde{f}(M) \\
&\quad - c(\widetilde{M}) \circ \widetilde{f}(\widetilde{M}) + \mathrm{diag}(c(\widetilde{M}) \circ \widetilde{f}(\widetilde{M}) \cdot \mathbf{1}_n)\widetilde{f}(\widetilde{M})\|_F \\
&\leq \|W\|_F \cdot \|X\|_F^2 \cdot (\|c(M) \circ \widetilde{f}(M) - c(\widetilde{M}) \circ \widetilde{f}(\widetilde{M})\|_F \\
&\quad + \|\mathrm{diag}(c(M) \circ \widetilde{f}(M) \cdot \mathbf{1}_n)\widetilde{f}(M) - \mathrm{diag}(c(\widetilde{M}) \circ \widetilde{f}(\widetilde{M}) \cdot \mathbf{1}_n)\widetilde{f}(\widetilde{M})\|_F) \\
&= R^3 \cdot (\|c(M) \circ \widetilde{f}(M) - c(\widetilde{M}) \circ \widetilde{f}(\widetilde{M})\|_F \\
&\quad + \|\mathrm{diag}(c(M) \circ \widetilde{f}(M) \cdot \mathbf{1}_n)\widetilde{f}(M) - \mathrm{diag}(c(\widetilde{M}) \circ \widetilde{f}(\widetilde{M}) \cdot \mathbf{1}_n)\widetilde{f}(\widetilde{M})\|_F)
\end{aligned}
$$

where the first step and the second step follow from Fact B.5, the third step follows from triangle inequality, and the fourth step follows from Assumption E.1.

Then we have

$$
\begin{aligned}
&R^3 \cdot (\|c(M) \circ \widetilde{f}(M) - c(\widetilde{M}) \circ \widetilde{f}(\widetilde{M})\|_F \\
&\quad + \|\mathrm{diag}(c(M) \circ \widetilde{f}(M) \cdot \mathbf{1}_n)\widetilde{f}(M) - \mathrm{diag}(c(\widetilde{M}) \circ \widetilde{f}(\widetilde{M}) \cdot \mathbf{1}_n)\widetilde{f}(\widetilde{M})\|_F) \\
&\leq R^3 \cdot (24dn^{7/2}R^3\|M - \widetilde{M}\|_F + 6dn^{3/2}R^3\|M - \widetilde{M}\|_F) \\
&\leq R^3 \cdot (30dn^{7/2}R^3\|M - \widetilde{M}\|_F) \\
&= 30dn^{7/2}R^6\|M - \widetilde{M}\|_F
\end{aligned} \tag{13}
$$

where the first step follows from Lemma F.5 and Lemma F.7, the second step follows from $n \geq 1$.

For the second term, we have

$$
\|\lambda(M - \widetilde{M})\|_F = \lambda\|M - \widetilde{M}\|_F \tag{14}
$$

which follows from Fact B.5.

Finally, we have

$$
\|\nabla_M \mathcal{L}(M) - \nabla_M \mathcal{L}(\widetilde{M})\|_F \leq (\lambda + 30dn^{7/2}R^6) \cdot \|M - \widetilde{M}\|_F
$$

which follows from Eq. (12), Eq. (13), and Eq. (14). $\qquad\square$

## G CONVERGENCE OF GRADIENT DESCENT

### G.1 HELPFUL STATEMENTS

Here, we present useful facts that we use to prove our convergence result.

**Fact G.1.** *We can show that for $a, b \in \mathbb{R}$*

- *Part 1.*

$$
\sqrt{a^2 + b^2} \geq \frac{|a| + |b|}{\sqrt{2}}
$$

- *Part 2. Suppose $|a| > |b|$*

$$
\sqrt{|a| - |b|} \geq \sqrt{|a|} - \sqrt{|b|}
$$

*Proof.* **Proof of Part 1.** Square both side of the inequality in **Part 1.**, we have

$$
\mathrm{LHS} = a^2 + b^2
$$

and

$$
\mathrm{RHS} = \frac{a^2 + 2|a| \cdot |b| + b^2}{2}.
$$

So we just need to prove

$$\begin{aligned}
\text{LHS} - \text{RHS} &= a^2 + b^2 - \frac{a^2 + 2|a| \cdot |b| + b^2}{2} \\
&= \frac{a^2 + b^2 - 2|a| \cdot |b|}{2} \\
&= \frac{(|a| - |b|)^2}{2} \\
&\geq 0
\end{aligned}$$

which is hold because for any $x \in \mathbb{R}$, $x^2 \geq 0$.

**Proof of Part 2.** Square both side of the inequality in **Part 2.**, we have

$$\text{LHS} = |a| - |b|$$

and

$$\text{RHS} = |a| + |b| - 2\sqrt{|a||b|}$$

So we just need to prove

$$\begin{aligned}
\text{LHS} - \text{RHS} &= |a| - |b| - |a| - |b| + 2\sqrt{|a||b|} \\
&= 2\sqrt{|a||b|} - 2|b| \\
&= 2\sqrt{|b|}(\sqrt{|a|} - \sqrt{|b|}) \\
&\geq 0
\end{aligned}$$

which is hold because $|a| > |b|$ and $|b| \geq 0$. $\qquad\square$

## G.2 LOWER BOUND ON FROBENIUS NORM

In this section, we present the lemma for the lower bound on the Frobenius norm.

**Lemma G.2.** *If the following conditions hold*

- *Let $B \in \mathbb{R}^{d \times d}$.*

- *Let $M \in [0, 1]^{d \times d}$.*

- *Let $\lambda \in [0, 1]$ be some constant.*

- *Suppose that $\|B\|_F \leq R$.*

*Then, we can show*

- *Part 1.*

$$\|B + \lambda M\|_F^2 \geq \|B\|_F^2 + \lambda^2 \|M\|_F^2 - 2R\lambda d$$

- *Part 2.*

$$\|B + \lambda M\|_F \geq \frac{1}{\sqrt{2}}(\|B\|_F + \lambda\|M\|_F) - \sqrt{2R\lambda d}$$

*Proof.* **Proof of Part 1.** We can show that

$$\begin{aligned}
\|B + \lambda M\|_F^2 &= \|B\|_F^2 + \lambda^2 \|M\|_F^2 + 2\langle B, \lambda M \rangle \\
&\geq \|B\|_F^2 + \lambda^2 \|M\|_F^2 - 2\|B\|_F \cdot \|\lambda M\|_F \\
&\geq \|B\|_F^2 + \lambda^2 \|M\|_F^2 - 2R\lambda d
\end{aligned} \tag{15}$$

where the first step follows from Fact B.6, the second step follows from Fact B.5, the third step follows from the upper bound of $\|B\|_F$ and $\|M\|_F$.

**Proof of Part 2.** Taking the square root on both sides, we get

$$\|B + \lambda M\|_F \geq \sqrt{\|B\|_F^2 + \lambda^2\|M\|_F^2 - 2R\lambda d}$$

$$\geq \sqrt{\|B\|_F^2 + \lambda^2\|M\|_F^2} - \sqrt{2R\lambda d}$$

$$\geq \frac{1}{\sqrt{2}}(\|B\|_F + \lambda\|M\|_F) - \sqrt{2R\lambda d}$$

where the first step follows from Eq. (15), the second step follows from **Part 2.** of Fact G.1, and the third step follows from **Part 1.** of Fact G.1.

□

### G.3 SANDWICH LOWER BOUND ON FROBENIUS NORM

Here, we introduce a sandwich trace fact.

**Fact G.3.** *If* $A \succeq \beta I$, *then* $\mathrm{tr}[B^\top AB] \geq \beta\,\mathrm{tr}[B^\top B]$.

*Proof.* As $A \succeq \beta I$, we have $A - \beta I \succeq 0$. Multiplying both sides by $B^\top$ on the left and $B$ on the right (noting that these operations preserve the positive semidefiniteness), we have

$$B^\top(A - \beta I)B \succeq 0.$$

Taking the trace and utilizing the property that the trace of a positive semidefinite matrix is non-negative, we have

$$\mathrm{tr}[B^\top AB - \beta B^\top B] \geq 0,$$

which simplifies to

$$\mathrm{tr}[B^\top AB] - \beta\,\mathrm{tr}[B^\top B] \geq 0.$$

This concludes the proof. □

We establish a sandwich lower bound on the Frobenius norm.

**Lemma G.4** (Formal version of Lemma 5.6). *If the following conditions hold*

- *Let* $B \in \mathbb{R}^{n \times n}$ *and* $X \in \mathbb{R}^{n \times d}$.

- *Assume that* $XX^\top \succeq \beta I$.

*Then, we have*

$$\|X^\top BX\|_F \geq \beta\|B\|_F$$

*Proof.* We can show that

$$\|X^\top BX\|_F^2 = \mathrm{tr}[X^\top BXX^\top B^\top X]$$

$$\geq \beta \cdot \mathrm{tr}[X^\top BB^\top X]$$

$$= \beta \cdot \mathrm{tr}[B^\top XX^\top B]$$

$$\geq \beta^2 \cdot \mathrm{tr}[B^\top B]$$

$$= \beta^2 \cdot \|B\|_F^2$$

where the first step, the third step, and the fifth step follow from Fact B.4, the second step and the fourth step follows from Fact G.3 and $XX^\top \succeq \beta I$.

Taking the square root of both sides, we finish the proof. □

## G.4 LOWER BOUND ON HADAMARD PRODUCT BETWEEN TWO MATRICES

We present the lemma for the lower bound on the Hadamard product between two matrices in this section.

**Lemma G.5.** *If the following conditions hold*

- *Let $B, W \in \mathbb{R}^{d \times d}$.*

*Then, we have*

$$\max_{i,j \in [d]} \{|W_{i,j}|\} \cdot \|B\|_F \geq \|W \circ B\|_F \geq \min_{i,j \in [d]} \{|W_{i,j}|\} \cdot \|B\|_F.$$

*Proof.* The proof directly follows from the definition of the Frobenius norm. □

## G.5 FINAL BOUND

We introduce some useful lemmas that we use to prove the final bound.

**Lemma G.6.** *If the following conditions hold*

- *Let $b \in \mathbb{R}^n$ and $\langle b, \mathbf{1}_n \rangle = 0$.*

- *Let $f \in [\delta, 1]^n$ and $\langle f, \mathbf{1}_n \rangle = 1$.*

*Then we have*

$$\|(b - \langle b, f \rangle \mathbf{1}_n) \circ f\|_2 \geq \delta \|b\|_2.$$

*Proof.* Note that $\langle b, \mathbf{1}_n \rangle = 0$ so that $b$ and $\mathbf{1}_n$ are orthogonal with each other. Then, we have

$$\|(b - \langle b, f \rangle \mathbf{1}_n) \circ f\|_2 \geq \delta \|b - \langle b, f \rangle \mathbf{1}_n\|_2$$
$$= \delta \sqrt{\|b\|_2^2 + \|\langle b, f \rangle \mathbf{1}_n\|_2^2}$$
$$\geq \delta \|b\|_2,$$

where the second step is from the Pythagorean theorem. □

We present our final bound for proving the PL inequality.

**Lemma G.7** (Formal version of Lemma 5.7). *If the following conditions hold*

- *Let $B \in \mathbb{R}^{n \times n}$ and each row summation is zero, i.e., $B \cdot \mathbf{1}_n = \mathbf{0}_n$.*

- *Let $\widetilde{f}(M) \in [0, 1]^{n \times n}$ and each row summation is 1, i.e., $\widetilde{f}(M) \cdot \mathbf{1}_n = \mathbf{1}_n$.*

- *Assume that $\min_{i,j \in [n]} \widetilde{f}(M)_{i,j} \geq \delta > 0$.*

*Then, we can show*

$$\|B \circ \widetilde{f}(M) - \operatorname{diag}((B \circ \widetilde{f}(M)) \cdot \mathbf{1}_n) \widetilde{f}(M)\|_F \geq \delta \cdot \|B\|_F$$

*Proof.* For any $i \in [n]$, let $B_i \in \mathbb{R}^n$ be the $i$-th row of $B$, and we have $\langle B_i, \mathbf{1}_n \rangle = 0$ by the first condition.

For any $i \in [n]$, let $\widetilde{f}(M)_i \in \mathbb{R}^n$ be the $i$-th row of $\widetilde{f}(M)$, and we have $\langle \widetilde{f}(M)_i, \mathbf{1}_n \rangle = 1$ by the second condition and $\widetilde{f}(M)_{i,j} \in [\delta, 1]$ by the third condition.

By Lemma G.6, for any $i \in [n]$, we have

$$\|(B_i - \langle B_i, \widetilde{f}(M)_i \rangle \mathbf{1}_n) \circ \widetilde{f}(M)_i\|_2 \geq \delta \|B_i\|_2.$$

Then, we have

$$\|B \circ \widetilde{f}(M) - \operatorname{diag}((B \circ \widetilde{f}(M)) \cdot \mathbf{1}_n) \widetilde{f}(M)\|_F^2$$

$$
\begin{aligned}
&= \sum_{i \in [n]} \|(B_i - \langle B_i, \widetilde{f}(M)_i \rangle \mathbf{1}_n) \circ \widetilde{f}(M)_i\|_2^2 \\
&\geq \sum_{i \in [n]} \delta^2 \|B_i\|_2^2 \\
&= \delta^2 \|B\|_F^2.
\end{aligned}
$$

$\square$

### G.6 PL Inequality

Here, we present the bound for one unit loss function.

**Lemma G.8.** *If the following conditions hold*

- *Let $c(M)$ be defined in Definition C.4.*

*We have*

$$
\|c(M)\|_F \leq 2\sqrt{n}.
$$

*Proof.* We have

$$
\begin{aligned}
\|c(M)\|_F &\leq \|\widetilde{f}(M)\|_F + \|f\|_F \\
&\leq 2\sqrt{n},
\end{aligned}
$$

where the first step follows from Definition C.4 and triangle inequality, the second step follows $x_1^2 + \cdots + x_n^2 \leq (x_1 + \cdots + x_n)^2$ when $x_i \geq 0$ for any $i \in [n]$. $\square$

We present the lemma to prove the PL inequality.

**Lemma G.9.** *If the following conditions hold*

- *Let $\widetilde{f}(M)$ be defined in Definition C.3.*
- *Let $c(M)$ be defined in Definition C.4.*

*We have*

$$
\|\operatorname{diag}((c(M) \circ \widetilde{f}(M)) \cdot \mathbf{1}_n) \widetilde{f}(M)\|_F \leq \sqrt{n}.
$$

*Proof.* We have

$$
\begin{aligned}
\|\operatorname{diag}((c(M) \circ \widetilde{f}(M)) \cdot \mathbf{1}_n) \widetilde{f}(M)\|_F &\leq \max_{i \in n}\{|(c(M)_i \circ \widetilde{f}(M)_i) \cdot \mathbf{1}_n|\} \cdot \|\widetilde{f}(M)\|_F \\
&\leq \|\widetilde{f}(M)\|_F \\
&\leq \sqrt{n},
\end{aligned}
$$

where the first step is by Frobenius norm definition and the second step follows from $\langle \widetilde{f}(M)_i, \mathbf{1}_n \rangle = 1$ and $c(M)_i \in [-1, 1]^n$ for any $i \in [n]$. $\square$

Finally, we can show the lemma for PL inequality.

**Lemma G.10** (PL inequality, formal version of 5.5)**.** *If the following conditions hold,*

- *Let $M \in [0, 1]^{d \times d}$.*
- *Let $\lambda \in [0, 1]$ be some constant.*
- *Assume that $XX^\top \succeq \beta I$.*
- *Assume that $\min_{i,j \in [n]} \widetilde{f}(M)_{i,j} \geq \delta > 0$.*

- *Let $\mathcal{L}(M)$ be defined in Definition C.7.*

*Furthermore,*

- *Let $\alpha = 2$.*
- *Let $\mu = 2\min_{i,j\in[d]}\{|W_{i,j}|\} \cdot \beta \cdot \delta$.*
- *Let $\xi = 12\sqrt{n}\max_{i,j\in[d]}\{|W_{i,j}|\} \cdot \|X\|_F^2 \cdot \lambda d/\mu$.*

*We have*

$$\|\nabla_M \mathcal{L}(M)\|_F^\alpha \geq \frac{1}{2}\mu(\|c(M)\|_F^2 + \frac{2\lambda^2}{\mu}\|M\|_F^2 - \xi).$$

*Proof.* We have $\widetilde{f}(M) \cdot \mathbf{1}_n = \mathbf{1}_n$ and $f \cdot \mathbf{1}_n = \mathbf{1}_n$ by Definition C.3. Note that $c(M) = \widetilde{f}(M) - f$ by Definition C.4. Thus, we have $c(M) \cdot \mathbf{1}_n = \mathbf{0}_n$.

On the other hand, we have

$$\|W \circ (X^\top(c(M) \circ \widetilde{f}(M) - \operatorname{diag}((c(M) \circ \widetilde{f}(M)) \cdot \mathbf{1}_n)\widetilde{f}(M))X)\|_F$$
$$\leq \max_{i,j\in[d]}\{|W_{i,j}|\} \cdot \|X^\top(c(M) \circ \widetilde{f}(M) - \operatorname{diag}((c(M) \circ \widetilde{f}(M)) \cdot \mathbf{1}_n)\widetilde{f}(M))X\|_F$$
$$\leq \max_{i,j\in[d]}\{|W_{i,j}|\} \cdot \|X\|_F^2 \cdot \|c(M) \circ \widetilde{f}(M) - \operatorname{diag}((c(M) \circ \widetilde{f}(M)) \cdot \mathbf{1}_n)\widetilde{f}(M)\|_F$$
$$\leq \max_{i,j\in[d]}\{|W_{i,j}|\} \cdot \|X\|_F^2 \cdot (\|c(M) \circ \widetilde{f}(M)\|_F + \|\operatorname{diag}((c(M) \circ \widetilde{f}(M)) \cdot \mathbf{1}_n)\widetilde{f}(M)\|_F)$$
$$\leq \max_{i,j\in[d]}\{|W_{i,j}|\} \cdot \|X\|_F^2 \cdot (\|c(M)\|_F + \|\operatorname{diag}((c(M) \circ \widetilde{f}(M)) \cdot \mathbf{1}_n)\widetilde{f}(M)\|_F)$$
$$\leq \max_{i,j\in[d]}\{|W_{i,j}|\} \cdot \|X\|_F^2 \cdot (2\sqrt{n} + \sqrt{n})$$
$$= \max_{i,j\in[d]}\{|W_{i,j}|\} \cdot \|X\|_F^2 \cdot 3\sqrt{n}$$

where the first and fourth steps follow Lemma G.5, the second step follows from Frobenius norm property, the third step follows from triangle inequality, the fifth step follows from Lemma G.8 and Lemma G.9.

Let $\alpha = 2$. We have the following

$$\|\nabla_M \mathcal{L}(M)\|_F^2$$
$$= \|W \circ (X^\top(c(M) \circ \widetilde{f}(M) - \operatorname{diag}((c(M) \circ \widetilde{f}(M)) \cdot \mathbf{1}_n)\widetilde{f}(M))X) + \lambda M\|_F^2$$
$$\geq \|W \circ (X^\top(c(M) \circ \widetilde{f}(M) - \operatorname{diag}((c(M) \circ \widetilde{f}(M)) \cdot \mathbf{1}_n)\widetilde{f}(M))X)\|_F^2 + \lambda^2\|M\|_F^2 - \alpha_1$$
$$\geq \alpha_2 \cdot \|X^\top(c(M) \circ \widetilde{f}(M) - \operatorname{diag}((c(M) \circ \widetilde{f}(M)) \cdot \mathbf{1}_n)\widetilde{f}(M))X\|_F^2 + \lambda^2\|M\|_F^2 - \alpha_1$$
$$\geq \alpha_2 \cdot \alpha_3 \cdot \|c(M) \circ \widetilde{f}(M) - \operatorname{diag}((c(M) \circ \widetilde{f}(M)) \cdot \mathbf{1}_n)\widetilde{f}(M)\|_F^2 + \lambda^2\|M\|_F^2 - \alpha_1$$
$$\geq \alpha_2 \cdot \alpha_3 \cdot \alpha_4 \cdot \|c(M)\|_F^2 + \lambda^2\|M\|_F^2 - \alpha_1$$
$$= \frac{1}{2}\mu(\|c(M)\|_F^2 + \frac{2\lambda^2}{\mu}\|M\|_F^2 - \xi),$$

where the second step follows from Lemma G.2 and $\alpha_1 = 6\sqrt{n}\max_{i,j\in[d]}\{|W_{i,j}|\} \cdot \|X\|_F^2 \cdot \lambda d$, the third step follows from Lemma G.5 and $\alpha_2 = \min_{i,j\in[d]}\{|W_{i,j}|\}$, the fourth step follows from Lemma G.4 and $\alpha_3 = \beta$, the fifth step follows from Lemma G.7 and $\alpha_4 = \delta$, and the last step follows from $\mu = 2\alpha_2 \cdot \alpha_3 \cdot \alpha_4$ and $\xi = 2\alpha_1/\mu$. □

