# OpenReview forum: "Beyond Linear Approximations: A Novel Pruning Approach for Attention Matrix"
_ICLR.cc/2025/Conference — ICLR 2025 Poster_

### Official Review · Reviewer_QLv4 · 2024-11-04

**Soundness:** 3
**Presentation:** 3
**Contribution:** 2
**Rating:** 6
**Confidence:** 2

**Summary:**

This paper proposes a method for unstructured pruning of the query and key projection matrices in attention. The paper provides theoretical guarantees for the convergence of the pruning method. It also provides some experiments on synthetic data demonstrating that the pruning method outperforms other pruning methods.

**Strengths:**

*Original problem.* The paper highlights an interesting, niche problem in the pruning literature, which, to my knowledge, has not been analyzed theoretically.

*Clear presentation.* The paper is well-written and the structure is easy to follow.

*Extensive theoretical analysis.* The authors provide an extensive theoretical analysis on the convergence of their pruning algorithm.

**Weaknesses:**

*Limited potential for speedup on large scale models.* I have two concerns around the potential for this method to provide significant speedups for large language models. (1) It is typically difficult to leverage fine-grained, unstructured sparsity for speedups on GPUs because standard vector and matrix instructions are not compatible with sparsity (though, recent changes to Nvidia GPUs could begin to mitigate this problem [Nvidia Blog](https://developer.nvidia.com/blog/accelerating-inference-with-sparsity-using-ampere-and-tensorrt/)). (2) The technique is applicable to a small fraction of the parameters in the full model. In large models, the MLPs dominate the total parameter and FLOP counts, but this method is only applicable to the Q and K projections, which account for only 9.4% of parameters in Llama-405B (excluding embeddings, [source](https://huggingface.co/meta-llama/Llama-3.1-405B/tree/main)). So even if the method could achieve 100% sparsity, the relative reduction in FLOPs/memory consumption would be limited.

*Experiments limited to synthetic data.* Because experiments are only performed on synthetic data and a single layer, it remains unclear how the performance of real language models would degrade with this method. Without experiments on real language models it is hard to assess the improvement of this method relative to prior work (Wanda and SparseGPT).

**Questions:**

How does the convergence rate in Theorem 4.1 compare to the rates one can achieve with naive baselines?

---

> ### Author Response · Authors · 2024-11-20
>
> ### W1.1: It is typically difficult to leverage fine-grained, unstructured sparsity for speedups on GPUs because standard vector and matrix instructions are not compatible with sparsity (though, recent changes to Nvidia GPUs could begin to mitigate this problem).
>
> Thanks for pointing out. Our work provides the foundation for future sparsity speedup. Specific sparsity patterns, e.g., structured N:M sparsity [1], and row/column/block-wise sparsity, are indeed important and widely used in modern GPU and CUDA implementation. However, the unstructured sparsity is a superset of specific sparsity patterns mentioned above, suggesting the potential algorithm design. To give a motivational example, we can modify our constraint to row-wise or column-wise pruning, which means we modify our pruning mask $M$ from a matrix of $d \times d$ to a vector of size $d$ denoting the index which row/column to prune, and then modify the corresponding loss function $\mathcal{L}$. This requires a re-analysis.
>
> ### W1.2: The technique is applicable to a small fraction of the parameters in the full model, because in large models, the MLPs dominate the total parameter and FLOP counts.
>
> Thanks for your valuable insights. Our method is applicable not only to attention layers but also to MLP layers, as we primarily employ gradient descent, which can be applied to the weights of MLP layers with tailored loss functions. Note that, MLP pruning is still in the linear approximation pruning regime, so it is not our main focus since their convergence is guaranteed.
>
> Furthermore, our approach can be seamlessly integrated with other linear pruning methods, such as Wanda [1] and SparseGPT [2], to prune MLP layers, achieving greater overall acceleration.
>
> ### W2: Experiments limited to synthetic data.
>
> Thank you for pointing out your concerns. We provide some results on the real dataset in C4 and the model in Llama 3.2 1B, where our method outperforms Wanda and SparseGPT. See global response **Experiments on real LLMs**.
>
> ### Q1: How does the convergence rate in Theorem 4.1 compare to the rates one can achieve with naive baselines?
>
> Thanks for your valuable question. Since pruning is typically performed only once, the associated time cost can be relatively high. However, after pruning, inference time is accelerated and unaffected by the initial pruning process. Wanda, being a non-iterative method, requires time equivalent to a single forward pass. In contrast, SparseGPT performs pruning through $d$ iterations, where $d$ represents the dimensionality.
>
> Our time cost of pruning mainly depends on the iterations which is $O(d \cdot \mathrm{poly}(n)/\epsilon)$. This is due to the relatively slow convergence of Gradient Descent. However, future work will explore second-order methods like Newton’s method or Adam optimizer and try to optimize the time and memory cost.
>
>
> [1] Sun, M., Liu, Z., Bair, A., & Kolter, J. Z. A Simple and Effective Pruning Approach for Large Language Models. ICLR 2024
>
> [2] Frantar, E., & Alistarh, D. Sparsegpt: Massive language models can be accurately pruned in one-shot. ICML 2023

---

> > ### Comment · Reviewer_QLv4 · 2024-11-25
> >
> > Thank you to the authors for their response and new experiments on real language models.
> >
> > To further strengthen the evaluation on LLMs in the next draft of the paper, I recommend that the authors actually measure end-to-end perplexity on C4 (not just the relative error for a layer's output).

---

> > > ### Author Response · Authors · 2024-11-25
> > > **Thank you**
> > >
> > > Thank you for your reply. We appreciate your time and feedback. As the time is very close to the end of the rebuttal period, we are not able to finish the new experiments suggested by the reviewer. We leave it as our future work.
> > >
> > > On the other hand, we would like to highlight this work is theoretically oriented, where we study a clean and well-established optimization problem, which is clearly abstracted from a concrete machine learning task. The theoretical part is our main focus and contribution, where we introduce a new approach to pruning the non-linear attention matrix and provide a convergence guarantee. Furthermore, our contribution is novel, even if we only consider the theoretical part. To the best of our knowledge, this is the first work studying non-linear pruning in transformers. We believe our work may inspire a wide range of new algorithm designs.
> > >
> > > Again, we thank you for the reviewers’ effort and constructive comments.

---

> > > > ### Author Response · Authors · 2024-11-26
> > > >
> > > > We agree with you and Reviewer opkW that end-to-end perplexity should be measured. As the rebuttal period extends, we’re working on that and probably can get some results given the extended rebuttal period. We will back to you later. Thank you again for your valuable suggestions!

---

> > > > > ### Author Response · Authors · 2024-11-29
> > > > > **Thank you and end-to-end perplexity evaluation**
> > > > >
> > > > > We thank the reviewers for their valuable suggestions. We appreciate your time and effort in the discussion. We evaluate our methods Perplexity (PPL) on a real model and real dataset from end to end, comparing Wanda [1] and SparseGPT [2]. We refer the reviewer to the new Global Response named **End-to-end Perplexity Evaluation** for details about the setup.
> > > > >
> > > > > The results are in the following table, where Dense means non-pruning/original weights.
> > > > >
> > > > > | Method | PPL |
> > > > > |------|----|
> > > > > | Dense MLP + Dense Attn | 12.487  |
> > > > > | Dense MLP + SparseGPT Attn | 14.269 |
> > > > > | Dense MLP + Wanda Attn | 14.912 |
> > > > > | Dense MLP + Our Attn | **13.885**  |
> > > > > | Wanda MLP + Wanda Attn | 30.426 |
> > > > > | Wanda MLP + SparseGPT Attn |  26.074 |
> > > > > | Wanda MLP + Our Attn | **24.427** |
> > > > > | SparseGPT MLP + Wanda Attn |45.435 |
> > > > > | SparseGPT MLP + SparseGPT Attn | 36.641  |
> > > > > | SparseGPT MLP + Our Attn | **34.946** |
> > > > >
> > > > > As we can see in the table, Our Attention pruning methods always outperform Wanda Attention pruning and SparseGPT Attention pruning by a large margin when combined with Dense/Wanda/SparseGPT MLP pruning methods.
> > > > > - These empirical results support our theoretical analysis that our pruning method can converge.
> > > > > - The empirical results support that our method is practical in the real world case.
> > > > > - Our method is broadly applicable and can be combined with many other pruning methods.
> > > > >
> > > > > We hope that our response can relieve your concerns. We are willing to discuss more if the reviewer has more follow-up questions.
> > > > >
> > > > > [1] Sun, M., Liu, Z., Bair, A., & Kolter, J. Z. A Simple and Effective Pruning Approach for Large Language Models. ICLR 2024
> > > > >
> > > > > [2] Frantar, E., & Alistarh, D. Sparsegpt: Massive language models can be accurately pruned in one-shot. ICML 2023

---

> > > > > > ### Author Response · Authors · 2024-12-02
> > > > > > **Looking forward to receiving your new feedback**
> > > > > >
> > > > > > Dear Reviewer QLv4,
> > > > > >
> > > > > > We hope we have adequately addressed your new concerns. We appreciate your valuable time. We would be very grateful if you could provide new feedback since the discussion deadline is approaching in several hours.
> > > > > >
> > > > > > If you require further clarification or have any additional concerns, please do not hesitate to contact us. We are more than willing to continue communicating with you.
> > > > > >
> > > > > > Warmest regards,
> > > > > >
> > > > > > Authors

---

> > > > > > > ### Comment · Reviewer_QLv4 · 2024-12-03
> > > > > > >
> > > > > > > Thanks to the authors for running these additional experiments within the rebuttal period. I think the end-to-end perplexity evaluations improve the paper and so I’m raising my score.
> > > > > > >
> > > > > > > For the camera ready submission, I encourage the authors to:
> > > > > > > - Specify in the table the model size (in parameters) and the number of tokens seen during training.
> > > > > > > - Specify the pretraining dataset (e.g. C4, Pile, Wikitext)
> > > > > > > - If computational resources and time permits, include results for larger models trained on more tokens.

---

> ### Author Response · Authors · 2024-12-03
> **Thank you**
>
> We appreciate your raising score!
>
> We use LLama 3.2 1B and C4 Dataset and we do not truncate the input sequences for new experiments. We will add these information and new empirical results in our camera ready. We will also add results of larger model and different datasets in camera ready.
>
> We sincerely thank you for your valuable suggestions and time!

---

### Official Review · Reviewer_N1Ar · 2024-11-05

**Soundness:** 3
**Presentation:** 3
**Contribution:** 3
**Rating:** 6
**Confidence:** 2

**Summary:**

The paper introduces a novel approach to pruning weights in large language models (LLMs) to improve computational efficiency, particularly for deployment on resource-constrained devices. This method focuses on optimizing the approximation of the attention matrix, a critical component in transformer architectures, by accounting for the non-linear properties of the Softmax attention mechanism. The authors present a Gradient Descent-based optimization technique, offering theoretical guarantees for its convergence toward a near-optimal pruning solution. Preliminary results indicate that this approach maintains model performance while significantly reducing computational costs. This work provides a new theoretical basis for designing pruning algorithms for LLMs, potentially enabling more efficient inference on devices with limited resources.

**Strengths:**

1. The presented technique enhances convergence guarantees for non-convex functions, crucial for optimization challenges in areas like deep learning, through the application of the Polyak-Łojasiewicz inequality. It showcases mathematical rigor with detailed derivations and validations of key conditions, supported by established theorems, notably Theorem 5.2 from Frei & Gu (2021). Practical implications include improved computational efficiency via attention weight pruning without notable performance trade-offs, assured by the method’s convergence mechanisms. Additionally, its framework offers potential applicability across various non-convex optimization scenarios, expanding its usefulness in the optimization toolkit.
2.  The experimental results suggest that the proposed algorithm outperforms existing methods under certain conditions, the above weaknesses highlight areas where the study could be improved.

**Weaknesses:**

1. The convergence guarantees depend on strict conditions like the boundedness and positive definiteness of matrices (e.g., $\( X^T X \succeq \beta I \))$. These conditions may not always hold in real-world applications, limiting the practical relevance of the results. Additionally, the analysis does not consider the stochasticity and noise common in real-world data and optimization, which could affect gradient descent’s performance.

2. The experimental section lacks clarity on whether the tested input sequence lengths cover the full range required for real-world corpus. Given that modern llm often deals with lengthy texts like entire documents or extended conversations, the study does not explore the algorithm’s performance in such scenarios, raising concerns about its practical utility.

**Questions:**

Conside the weakness

---

> ### Author Response · Authors · 2024-11-20
>
> ### W1: Assumed conditions may not always hold in real-world applications, limiting the practical relevance of the results.
>
> Thank you for pointing out your concerns. We provide some results to verify the assumptions on the real dataset in C4 and the model in Llama 3.2 1B. See global response **Assumptions verification on real LLMs**.
>
> ### W2: The experimental section lacks clarity on whether the tested input sequence lengths cover the full range required for real-world corpus. Given that modern llm often deals with lengthy texts like entire documents or extended conversations, the study does not explore the algorithm’s performance in such scenarios, raising concerns about its practical utility.
>
> Thank you for pointing out. We provide some results on the real dataset in C4 and model in Llama 3.2 1B for a variation of input context length $n$. See global response **Experiments on real LLMs**.

---

> > ### Author Response · Authors · 2024-11-29
> > **Thank you and end-to-end perplexity evaluation**
> >
> > Per reviewers QLv4 and opkW's valuable suggestions, we evaluate our methods Perplexity (PPL) on a real model and real dataset from end to end, comparing Wanda [1] and SparseGPT [2]. We refer the reviewer to the new Global Response named **End-to-end Perplexity Evaluation** for details about setup and results.
> >
> > As we can see in the table, Our Attention pruning methods always outperform Wanda Attention pruning and SparseGPT Attention pruning by a large margin when combined with Dense/Wanda/SparseGPT MLP pruning methods.
> > - These empirical results support our theoretical analysis that our pruning method can converge.
> > - The empirical results support that our method is practical in the real world case.
> > - Our method is broadly applicable and can be combined with many other pruning methods.
> >
> > We are willing to discuss more if the reviewer has more follow-up questions.
> >
> > [1] Sun, M., Liu, Z., Bair, A., & Kolter, J. Z. A Simple and Effective Pruning Approach for Large Language Models. ICLR 2024
> >
> > [2] Frantar, E., & Alistarh, D. Sparsegpt: Massive language models can be accurately pruned in one-shot. ICML 2023

---

### Official Review · Reviewer_WRvG · 2024-11-05

**Soundness:** 2
**Presentation:** 3
**Contribution:** 3
**Rating:** 6
**Confidence:** 4

**Summary:**

The paper proposes to prune the projection weight matrices of the Softmax attention conditioned on the similarity between the pruned and the original attention matrices. The proposed approach leverages gradient decent to solve the constraint opimitzation problem and has theoretical gurantees for near-optimality with polynormal time.

**Strengths:**

1. Considering the similarity between attention matrices during pruning of the projection weight is novel and interesting.
2. The assumptions in theoretical analyses are mild and sensible.

**Weaknesses:**

1. The applicability of the approach is limited. The proposed pruning approach needs to fuse the query and the key projection matrices into a single matrix through matrix multiplication. This means that it will eliminate the inference efficiency benefits of popular LLMs, such as Llama 3, which use Grouped Query Attention that shares the key-value pairs between queries. This is because after the proposed fushion, the original input X needs to be saved during decoding, while for GQA, only much smaller key-value vectors are needed which has much smaller memory I/O footprint.
2. Lacks of empirical results on pruning real-world LLM. The low error rates on synthetic data may not translate to real world dowstream performances.

**Questions:**

1. How to revert the pruned fused matrix back to the query and key projection matrices respectively?
2. How does the proposed attention pruning affect length extrapolation ability?

---

> ### Author Response · Authors · 2024-11-20
>
> ### W1: The applicability of the approach is limited. The proposed pruning approach needs to fuse the query and the key projection matrices into a single matrix through matrix multiplication. This means that it will eliminate the inference efficiency benefits of popular LLMs, such as Llama 3, which use Grouped Query Attention that shares the key-value pairs between queries. This is because after the proposed fusion, the original input X needs to be saved during decoding, while for GQA, only much smaller key-value vectors are needed which has much smaller memory I/O footprint.
>
> Thanks for your insightful comments. Our methods can easily extend to GQA setting.
>
> In our new experiments in global response **Experiments on real LLMs**, we add two pruning masks to $W_K$ and $W_Q$ separately, where the two masks are the optimization variables, and the objective/loss function is calculated based on these two masks (see Line 479-502 in revision). Thus, when we consider GQA, we can use joint objective function over all attention heads in a group of GQA. For example, Llama 3.2 1B has "num_attention_heads": 32, "num_key_value_heads": 8, and “num_key_value_groups”: 4. Thus, we can use 4 pruning masks for 4 $W_Q$ and 1 pruning mask for 1 $W_K$ in this group as the optimization variables, and, correspondingly, the overall objective/loss function is the summation of 4 loss functions over 4 attention heads.
>
> There is a small gap between our current analysis setup and the GQA extension. However, this small gap may not hurt the theoretical intuition of our non-linear approximation pruning. We are willing to discuss more per the reviewer’s request.
>
> ### W2: Lacks of empirical results on pruning real-world LLM. The low error rates on synthetic data may not translate to real world dowstream performances.
>
> Thank you for pointing out your concerns. We provide some results on the real dataset in C4 and the model in Llama 3.2 1B, where our method outperforms Wanda and SparseGPT. See global response **Experiments on real LLMs** for details.
>
>
> ### Q1: How to revert the pruned fused matrix back to the query and key projection matrices respectively?
> We can add two pruning masks to $W_K$ and $W_Q$, and let them be optimization variables as discussed in **W1** above. In this way, we can get query and key projection matrices respectively.
>
> ### Q2: How does the proposed attention pruning affect length extrapolation ability?
> Thank you for your insightful question. The length extrapolation ability is a very important feature for LLMs. But we have not forsee an answer based on our current analysis directly. We will keep it in mind and leave this important topic as our future work.

---

> > ### Author Response · Authors · 2024-11-29
> > **Thank you and end-to-end perplexity evaluation**
> >
> > Per reviewers QLv4 and opkW's valuable suggestions, we evaluate our methods Perplexity (PPL) on a real model and real dataset from end to end, comparing Wanda [1] and SparseGPT [2]. We refer the reviewer to the new Global Response named **End-to-end Perplexity Evaluation** for details about setup and results.
> >
> > As we can see in the table, Our Attention pruning methods always outperform Wanda Attention pruning and SparseGPT Attention pruning by a large margin when combined with Dense/Wanda/SparseGPT MLP pruning methods.
> > - These empirical results support our theoretical analysis that our pruning method can converge.
> > - The empirical results support that our method is practical in the real world case.
> > - Our method is broadly applicable and can be combined with many other pruning methods.
> >
> > We are willing to discuss more if the reviewer has more follow-up questions.
> >
> > [1] Sun, M., Liu, Z., Bair, A., & Kolter, J. Z. A Simple and Effective Pruning Approach for Large Language Models. ICLR 2024
> >
> > [2] Frantar, E., & Alistarh, D. Sparsegpt: Massive language models can be accurately pruned in one-shot. ICML 2023

---

> > > ### Comment · Reviewer_WRvG · 2024-11-30
> > >
> > > Thanks for the impressive evaluation results! I have raised my score to 6.

---

> > > > ### Author Response · Authors · 2024-11-30
> > > > **Thank you**
> > > >
> > > > We are glad that our response fixed your concerns. We thank the reviewer for their insightful comments. We appreciate your time and score raising!

---

### Official Review · Reviewer_opkW · 2024-11-05

**Soundness:** 2
**Presentation:** 3
**Contribution:** 2
**Rating:** 5
**Confidence:** 3

**Summary:**

This paper proposes a novel approach to pruning QK projection matrices by directly optimizing for an approximation of the attention matrix, bypassing traditional linear approximations applied before the Softmax operation. By addressing the non-linear nature of the attention mechanism, the authors introduce a gradient descent-based optimization method with theoretical guarantees on convergence, aiming to produce a near-optimal pruning mask. Preliminary empirical results, although limited, demonstrate that this approach can maintain model performance while significantly reducing computational costs.

**Strengths:**

- The paper offers a robust theoretical analysis, including convergence guarantees for the gradient descent algorithm, which enhances confidence in the method’s foundational soundness.
- The proposed pruning method consistently outperforms baseline approaches in terms of relative error, effectively recovering the attention matrix activations. This suggests its potential for maintaining model performance while reducing computation.

**Weaknesses:**

- The empirical analysis relies heavily on synthetic data, specifically a full-rank Gaussian random matrix, with matrix dimensions (64x128) that are small relative to the scale of practical LLMs. This somewhat limits confidence in the method’s real-world applicability.
- I understand the main contributions are theoretical, but I think the paper would benefit from experiments with actual LLM activations and weights, even small-scale LLMs. Results from real models would give a stronger sense of how effective this method might be in practice.
- One big limitation is that the approach only targets $W_Q$ and $W_K$ matrices in the attention mechanism, while the majority of parameters in transformers come from the MLP layers. It would be interesting to see if this technique could be extended to cover the MLP parameters as well, which could have a more substantial impact on reducing model size.

**Questions:**

- I assume the cost of pruning itself is very small given the closed-form gradient, but it would be helpful to see an analysis of this cost. Is this approach lightweight enough for large models?
- The assumptions regarding the Lipschitz continuity and PL inequality of the gradient are central to the theoretical guarantees. Are there empirical checks or practical insights into how these assumptions hold up in real LLMs?
- The paper uses synthetic data by generating $W_Q$ and $W_K$ through SVD on Gaussian random matrices. A bit more explanation for this choice—or testing on real model weights—would clarify whether the setup could translate to practical applications.

---

> ### Author Response · Authors · 2024-11-20
>
> ### W1 & W2 & Q3: Experiments with real LLMs and real data
>
> Thank you for pointing out your concern. We provide some results on the real dataset in C4 and the model in Llama 3.2 1B, where our method outperforms Wanda and SparseGPT. See global response **Experiments on real LLMs** for all details.
>
> ### W3: It would be interesting to see if this technique could be extended to cover the MLP parameters as well, which could have a more substantial impact on reducing model size.
>
> Thanks for your valuable suggestions. Our method is applicable not only to attention layers but also to MLP layers, as we primarily employ gradient descent, which can be applied to the weights of MLP layers with tailored loss functions. Note that, MLP pruning is still in the linear approximation pruning regime, so it is not our main focus since their convergence is guaranteed.
>
> Furthermore, our approach can be seamlessly integrated with other linear pruning methods, such as Wanda [1] and SparseGPT [2], to prune MLP layers, achieving greater overall acceleration.
>
> ### Q1: I assume the cost of pruning itself is very small given the closed-form gradient, but it would be helpful to see an analysis of this cost. Is this approach lightweight enough for large models?
>
> Thanks for your suggestions. The memory cost is not big since we have the close form of the gradient. The time cost of pruning mainly depends on the iterations which is $O(d \cdot \mathrm{poly}(n)/\epsilon)$. This is due to the relatively slow convergence of Gradient Descent. However, future work will explore second-order methods like Newton’s method or Adam optimizer and try to optimize the time and memory cost.
>
> ### Q2: The assumptions regarding the Lipschitz continuity and PL inequality of the gradient are central to the theoretical guarantees. Are there empirical checks or practical insights into how these assumptions hold up in real LLMs?
>
> Thanks for pointing out. Yes, we check our assumptions on Llama 3.2 1B. See global response **Assumptions verification on real LLMs**.
>
> [1] Sun, M., Liu, Z., Bair, A., & Kolter, J. Z. A Simple and Effective Pruning Approach for Large Language Models. ICLR 2024
>
> [2] Frantar, E., & Alistarh, D. Sparsegpt: Massive language models can be accurately pruned in one-shot. ICML 2023

---

> > ### Comment · Reviewer_opkW · 2024-11-25
> >
> > Thank you to the authors for the clarification and the new experiments with real language models.
> >
> > That said, I agree with Reviewer QLv4’s point that end-to-end perplexity should be measured. I’d also suggest including the ratio of pruned weights to the whole model. Since pruning is a practical tool and there are many ways to optimize a language model to reach similar efficiency, it would be helpful to show at least one end-to-end result, even if the paper is mainly theory-focused.
> >
> > For these reasons, I maintain my current score.

---

> > > ### Author Response · Authors · 2024-11-26
> > >
> > > Thank you for your reply. We appreciate your time and feedback.
> > >
> > > We agree with you and Reviewer QLv4 that end-to-end perplexity should be measured. As the rebuttal period extends, we’re working on that and probably can get some results given the extended rebuttal period. We will back to you later. Thank you again for your valuable suggestions!

---

> > > > ### Author Response · Authors · 2024-11-29
> > > > **Thank you and end-to-end perplexity evaluation**
> > > >
> > > > We thank the reviewers for their valuable suggestions. We appreciate your time and effort in the discussion. We evaluate our methods Perplexity (PPL) on a real model and real dataset from end to end, comparing Wanda [1] and SparseGPT [2]. We refer the reviewer to the new Global Response named **End-to-end Perplexity Evaluation** for details about the setup.
> > > >
> > > > The results are in the following table, where Dense means non-pruning/original weights.
> > > >
> > > > | Method | PPL |
> > > > |------|----|
> > > > | Dense MLP + Dense Attn | 12.487  |
> > > > | Dense MLP + SparseGPT Attn | 14.269 |
> > > > | Dense MLP + Wanda Attn | 14.912 |
> > > > | Dense MLP + Our Attn | **13.885**  |
> > > > | Wanda MLP + Wanda Attn | 30.426 |
> > > > | Wanda MLP + SparseGPT Attn |  26.074 |
> > > > | Wanda MLP + Our Attn | **24.427** |
> > > > | SparseGPT MLP + Wanda Attn |45.435 |
> > > > | SparseGPT MLP + SparseGPT Attn | 36.641  |
> > > > | SparseGPT MLP + Our Attn | **34.946** |
> > > >
> > > > As we can see in the table, Our Attention pruning methods always outperform Wanda Attention pruning and SparseGPT Attention pruning by a large margin when combined with Dense/Wanda/SparseGPT MLP pruning methods.
> > > > - These empirical results support our theoretical analysis that our pruning method can converge.
> > > > - The empirical results support that our method is practical in the real world case.
> > > > - Our method is broadly applicable and can be combined with many other pruning methods.
> > > >
> > > > We hope that our response can relieve your concerns. We are willing to discuss more if the reviewer has more follow-up questions.
> > > >
> > > > [1] Sun, M., Liu, Z., Bair, A., & Kolter, J. Z. A Simple and Effective Pruning Approach for Large Language Models. ICLR 2024
> > > >
> > > > [2] Frantar, E., & Alistarh, D. Sparsegpt: Massive language models can be accurately pruned in one-shot. ICML 2023

---

> > > > > ### Author Response · Authors · 2024-12-02
> > > > > **Looking forward to receiving your new feedback**
> > > > >
> > > > > Dear Reviewer opkW,
> > > > >
> > > > > We hope we have adequately addressed your new concerns. We appreciate your valuable time. We would be very grateful if you could provide new feedback since the discussion deadline is approaching in several hours.
> > > > >
> > > > > If you require further clarification or have any additional concerns, please do not hesitate to contact us. We are more than willing to continue communicating with you.
> > > > >
> > > > > Warmest regards,
> > > > >
> > > > > Authors

---

### Author Response · Authors · 2024-11-20
**Global Response**

Thanks for the reviews' valuable comments!

We appreciate reviewers opkW, N1Ar, WRvG, and QLv4 for recognizing the novelty and theoretical rigor of our analysis. They collectively acknowledge our convergence guarantees for the gradient descent method, with opkW, N1Ar, and QLv4 highlighting the lower relative error achieved compared to existing pruning methods. WRvG commends the mild and sensible assumptions in our theoretical analysis, while N1Ar notes our method’s potential applicability to various non-convex optimization scenarios, enhancing versatility. QLv4 appreciates the clarity and organization of our paper. Additionally, WRvG and QLv4 find our optimization approach to attention matrix pruning particularly intriguing.

We have updated a **revision** for our draft. We also updated the code about experiments in the supplemental material under the folder named *7932_supp_rebuttal*. We summarize all the updates (in brown color) we made in the revision. All line numbers in the rebuttal correspond to the revised version.
- Line 472-530: We add our new experimental results performed on Llama 3.2 1B and C4 dataset.

Here, we address some common questions.

## Experiments on real LLMs
We update the below experiments in our revision Line 472-530.

Due to the time constraint, we implement our method on [LLama 3.2 1B](https://huggingface.co/meta-llama/Llama-3.2-1B-Instruct/blob/main/config.json). We extract the real attention weights and input hidden states for one specific attention layer using a customized hook function and use the calibration data from [C4 Dataset](https://huggingface.co/datasets/legacy-datasets/c4) as Wanda [1] and SparseGPT [2]. We also submit our code in the supplementary material. We show our results below, where the number in each cell is the output **relative error** (Line 418-421, Equation (1) in the draft) between the pruned and original attention layers:

All three methods under different input context length $n$:
| **Method \ $n$** | **64**       | **128**      | **256**      | **512**      | **1024**     | **2048**     | **4096**     |
|---------------|--------------|--------------|--------------|--------------|--------------|--------------|--------------|
| **Wanda**     | 0.1146       | 0.1204       | 0.1271       | 0.1229       | 0.1533       | 0.1851       | 0.1804       |
| **Sparse GPT**| 0.0595       | 0.0692       | 0.0873       | 0.0749       | 0.0968       | 0.1526       | 0.1472       |
| **Ours** | 0.0576       | 0.0399       | 0.0542       | 0.0471       | 0.0847       | 0.1262       | 0.1346       |

All three methods under different pruning ratios $\rho$:
| Method \ $\rho$         | 0.4      | 0.5      | 0.6      | 0.7      | 0.8      |
|-----------------|--------------|--------------|--------------|--------------|--------------|
| **Wanda**| 0.0337       | 0.0491       | 0.0750       | 0.1159       | 0.1730       |
| **SparseGPT**| 0.0331      | 0.0329       | 0.0476       | 0.0763       | 0.0989       |
| **Ours**   | 0.0333       | 0.0333       | 0.0390       | 0.0423       | 0.0485       |


As we can see that our result is quite promising, where our method outperforms Wanda and SparseGPT in all cases. These results show that our method may inspire new pruning algorithms, and the current pruning techniques used for LLMs have potential improvement space. Lastly, we emphasize that our work is theoretical focus, introducing a new approach to pruning the non-linear attention matrix. We are actively researching ways to extend our pruning method to larger LLMs to show more promising empirical results.


## Assumptions verification on real LLMs

Our work made two assumptions:
- We assume the positive semi-definiteness of input data, i.e., $XX^\top \succeq \beta I$ in Line 256.
- We assume the pruned attention matrix is lower bounded, i.e. $\min_{i,j \in [n]} (\widetilde{D}^{-1}\widetilde{A})_{i,j} \ge \delta > 0$ in Line 256.

We verify the two assumptions on Llama 3.2 1B described in the table below:
| **Assumption**         | **Value**           |
|-------------------------|---------------------|
| PSD input data, $\beta$ assumption  | 0.034 |
| Attention matrix lower bound, $\delta$ assumption | 0.0025 |


We see our assumptions hold for Llama 3.2 1b on the C4 dataset, suggesting the practicality of our assumptions.

[1] Sun, M., Liu, Z., Bair, A., & Kolter, J. Z. A Simple and Effective Pruning Approach for Large Language Models. ICLR 2024

[2] Frantar, E., & Alistarh, D. Sparsegpt: Massive language models can be accurately pruned in one-shot. ICML 2023

---

### Author Response · Authors · 2024-11-29
**End-to-end Perplexity Evaluation**

Per reviewers QLv4 and opkW's valuable suggestions, we evaluate our methods Perplexity (PPL) on a real model and real dataset from end to end, comparing Wanda [1] and SparseGPT [2].

*Setup:* We use LLama 3.2 1B as the target model to prune, in which the hidden state dimension is 2048. We set the pruning ratio as 0.5 for MLP and 0.5 for Attention Layer. If we prune the MLP and Attention Layer, we have a pruning ratio of 0.5 for the whole model. We use 8 sentences in the C4 training dataset [3] as calibration data, we use 128 sentences in the C4 validation dataset [3] to evaluate perplexity. We set 0.005 as the learning rate and 0.05 as the regularization coefficient.

The results are in the following table, where Dense means non-pruning/original weights.

| Method | PPL |
|------|----|
| Dense MLP + Dense Attn | 12.487  |
| Dense MLP + SparseGPT Attn | 14.269 |
| Dense MLP + Wanda Attn | 14.912 |
| Dense MLP + Our Attn | **13.885**  |
| Wanda MLP + Wanda Attn | 30.426 |
| Wanda MLP + SparseGPT Attn |  26.074 |
| Wanda MLP + Our Attn | **24.427** |
| SparseGPT MLP + Wanda Attn |45.435 |
| SparseGPT MLP + SparseGPT Attn | 36.641  |
| SparseGPT MLP + Our Attn | **34.946** |

As we can see in the table, Our Attention pruning methods always outperform Wanda Attention pruning and SparseGPT Attention pruning by a large margin when combined with Dense/Wanda/SparseGPT MLP pruning methods.
- These empirical results support our theoretical analysis that our pruning method can converge.
- The empirical results support that our method is practical in the real world case.
- Our method is broadly applicable and can be combined with many other pruning methods.

[1] Sun, M., Liu, Z., Bair, A., & Kolter, J. Z. A Simple and Effective Pruning Approach for Large Language Models. ICLR 2024

[2] Frantar, E., & Alistarh, D. Sparsegpt: Massive language models can be accurately pruned in one-shot. ICML 2023

[3] Raffel, C., Shazeer, N., Roberts, A., Lee, K., Narang, S., Matena, M., ... & Liu, P. J. Exploring the limits of transfer learning with a unified text-to-text transformer. JMLR’20.

---

### Meta-Review · Area_Chair_Tuzm · 2024-12-20

**Metareview:**

This paper introduces a method for pruning query and key weight matrices in attention modules. The paper proposes a theoretical justification for the proposed scheme, with convergence guarantees. The method was initially evaluated on synthetic data, and as requested by the reviewers, also on real-world language models. The proposed approach seems to perform relatively well, but does not provide a viable strategy for pruning in real-world applications. As mentioned by reviewer opkW, most of the computation and parameters are in the feedforward layers. Tu summarize, this paper proposes an interesting finding, for which the empirical implications are yet a bit unclear. While this is a borderline paper, I recommend this paper for acceptance.

**Additional Comments On Reviewer Discussion:**

Reviewers enjoyed that the authors provided real-world evaluations as part of the rebuttal, with two reviewers increasing their scores from 5->6. This results in a much improved rating set : 5556 -> 5666.

---

### Decision · Program_Chairs · 2025-01-22

Accept (Poster)